# Aerosols overtake greenhouse gases causing a warmer climate and more weather extremes toward carbon neutrality

Pinya Wang [1], Yang Yang [1] ✉, Daokai Xue [2], Lili Ren[3], Jianping Tang[2], L. Ruby Leung [4] & Hong Liao [1]

To mitigate climate warming, many countries have committed to achieve carbon neutrality in the mid-21st century. Here, we assess the global impacts of changing greenhouse gases (GHGs), aerosols, and tropospheric ozone ($O_3$) following a carbon neutrality pathway on climate and extreme weather events individually using the Community Earth System Model version 1 (CESM1). The results suggest that the future aerosol reductions significantly contribute to climate warming and increase the frequency and intensity of extreme weathers toward carbon neutrality and aerosol impacts far outweigh those of GHGs and tropospheric $O_3$. It reverses the knowledge that the changing GHGs dominate the future climate changes as predicted in the middle of the road pathway. Therefore, substantial reductions in GHGs and tropospheric $O_3$ are necessary to reach the 1.5 °C warming target and mitigate the harmful effects of concomitant aerosol reductions on climate and extreme weather events under carbon neutrality in the future.

Due to human activities, atmospheric greenhouse gas (GHG) concentrations have increased dramatically since around 1750, causing the climate to warm in the last few hundred years. The rate of global warming has accelerated in recent decades, with the global mean surface air temperature rising by 1.1 °C above the pre-industrial levels[1]. Global warming increases the thermal and latent energy that boosts extreme weather events[2], such as heat waves, torrential rains, and flooding. For instance, the frequency, duration, and intensity of heat waves have been increasing worldwide in the past decades[3], which adversely affected ecosystems, agricultural systems, and economies[4].

Carbon emission from fossil fuel-based energy consumption is a primary cause of the global greenhouse effect. To minimize the negative impacts and risks of climate change, more than 110 countries have committed to achieving carbon neutrality in the mid-21st century[5–7]. Carbon neutrality is believed to be critical and urgently needed to meet the Paris Agreement's warming target of limiting

global warming to within 2 °C, but preferably to 1.5 °C by 2100[8]. Specifically, the United States rejoined the Paris Climate Agreement in 2021, pledging to achieve net-zero emissions of carbon dioxide ($CO_2$) by 2050, through carbon reduction programs by the federal and local government. For example, California passed Assembly Bill No. 32 (AB32) in 2006, proposing to cut carbon emissions to the 1990 levels by 2020 and to 80% lower levels by 2050[9]. The European Union's 27 member countries have promised to cut at least 55% of carbon emissions from the 1990 levels by 2030 and achieve carbon neutrality by 2050[10]. The Chinese government has established its ambitious goal of reaching a carbon peak by 2030 and carbon neutrality by 2060, endeavoring to gradually achieve net-zero carbon emissions[11].

Climate change and air pollution are two sides of the same coin. Extraction and burning of fossil fuels emit GHGs and other pollutants, including particulate matter (PM) and ozone ($O_3$) precursors, etc. Therefore, carbon neutrality would substantially reduce air pollution

[1]Jiangsu Key Laboratory of Atmospheric Environment Monitoring and Pollution Control, Jiangsu Collaborative Innovation Center of Atmospheric Environment and Equipment Technology, Joint International Research Laboratory of Climate and Environment Change, School of Environmental Science and Engineering, Nanjing University of Information Science and Technology, Nanjing, Jiangsu, China. [2]School of Atmospheric Sciences, Nanjing University, Nanjing, Jiangsu, China. [3]College of Environment and Ecology, Jiangsu Open University, Nanjing, Jiangsu, China. [4]Atmospheric Sciences and Global Change Division, Pacific Northwest National Laboratory, Richland, WA, USA. ✉e-mail: yang.yang@nuist.edu.cn

emissions, with co-benefits for air quality. The low-carbon policies driven by the carbon neutrality target of China are expected to reduce the emissions of sulfur dioxide ($SO_2$), nitrogen oxides ($NO_x$), primary $PM_{2.5}$ (particulate matter with an aerodynamic diameter <2.5 μm), and volatile organic compounds (VOCs) by 42%, 42%, 44%, and 28% in 2030 and by 93%, 93%, 90%, and 61% in 2060 respectively[12]. Consequently, the $PM_{2.5}$ and $O_3$–8h (the maximum 8-h moving average of $O_3$ concentration) 90th percentile concentrations are estimated to be reduced by more than 80% in 2060 relative to the 2020 levels, and all municipal cities will reach the current national air quality standard[12]. Compared to other emission reduction pathways, Cheng et al.[7] suggest that carbon neutrality pathways are necessary and effective if China is to meet international World Meteorological Organization (WMO) standards (i.e., annual mean less than 35 μg/m³) for $PM_{2.5}$ concentrations. In the United States, the multiple feasible carbon neutrality pathways such as producing low-carbon fuels from biomass and electrification would effectively reduce the emissions of air pollutants[13].

Besides reducing the co-emitted air pollutants, global GHG mitigations can also reduce air pollution by slowing climate change which alters the meteorology conducive to pollution production and/ or accumulations[14–18]. In turn, changes in air pollutants and/or their precursors can also affect the climate. Specifically, tropospheric $O_3$ in the atmosphere has a positive radiative forcing[19], while different components of PM can have either warming or cooling effects on the climate[20,21]. For example, black carbon (BC), a particulate pollutant from combustion, contributes to the warming of the climate, while sulfate aerosol cools the earth[22]. Therefore, reducing air pollution emissions toward carbon neutrality may dampen or boost the effects of GHG mitigations. Although multiple attempts have been made to understand how mitigation options affect climate change, previous studies mainly focused on changes in the mean climate, but changes in extreme weather events have more direct, immediate, and visible impacts on human health and socioeconomics[23]. Yang et al.[21] attributed the 2020 record-breaking rainfall in China partly to the abrupt emission reductions during the COVID-19 pandemic. It has also been argued that the increased surface air temperature and decreased relative humidity caused by the aerosol emission reductions during the pandemic explain one-third of the observed increase in wildfires during August–November over the western United States in 2020[24].

As part of the Coupled Model Intercomparison Project Phase 6 (CMIP6), the Detection and Attribution Model Intercomparison Project (DAMIP) has been conducted to estimate the individual contributions of various external forcings to the observed global and regional climate changes[25,26]. The experimental design includes historical and future simulations forced with GHG-only, aerosols-only, stratospheric-ozone-only, and natural-only forcing. However, the future experiments in DAMIP are conducted only under the Medium Shared Socioeconomic Pathway 2-4.5 (SSP2-4.5) scenario, representing an intermediate level of future changes in GHGs, aerosols, and land use. In this study, with a global-scale carbon neutrality pathway, i.e., the latest Shared Socioeconomic Pathway 1-1.9 (SSP1-1.9) scenario in CMIP6 which limits warming to 1.5 °C set forth by the Paris Agreement[27,28], we systematically investigate the impacts of the ambitious mitigation strategies on climate and extreme weather events, including extreme heat and precipitation. More specifically, we simulate the individual impacts attributed to changes in GHGs, aerosols, and tropospheric $O_3$ under the carbon neutrality scenario.

## Results

### Future changes in GHGs, aerosols, and ozone levels under carbon neutrality scenario
The global mean concentrations of GHGs for 2020, 2050, and 2100 under the carbon neutrality scenario are listed in Table S1. Specifically, the global mean concentration of $CO_2$ will increase from the current

level of 414 ppm to 437 ppm by 2050 and then decrease to 400 ppm by 2100 following the SSP1-1.9 pathway. The global mean concentration of methane ($CH_4$) will decrease from the current level of 1884 ppb to 1429 ppb by 2050, and further decrease to 1061 ppb by 2100, while nitrous oxide ($N_2O$) will increase from the current level of 332 ppb to 344 ppb and 353 ppb by 2050 and 2100, respectively.

In general, the tropospheric $O_3$ levels are projected to decrease across the globe by the mid-of-century, with apparent negative anomalies in tropospheric column $O_3$ (TCO) in 2050 and 2100 compared to the 2020 levels (Fig. S1b and 1c). The decreases in TCO are higher, exceeding −6 DU in 2050 and −10 DU in 2100, over mid-to-high latitudes of the Northern Hemisphere where the current levels of TCO are relatively high (Fig. S1a). The changes in surface $O_3$ concentrations agree with the TCO changes, with marked decreases above −10 ppb in 2050 and −20 ppb in 2100 over mid-to-high latitudes of the Northern Hemisphere, compared to 2020 levels (Fig. S1d–f).

In pace with the strict control of anthropogenic emissions under carbon neutrality, the emissions of aerosols and their precursors will decrease significantly in the future. Specifically, the changes in emissions of $SO_2$ are mostly concentrated over East Asia, with the emission decreasing from 3 g m⁻² a⁻¹ to lower than 1.4 g m⁻² a⁻¹ in 2050 and lower than 1.0 g m⁻² a⁻¹ in 2100 (Fig. S2a –c). And the emissions of BC over the polluted regions such as East Asia, South Asia, Europe, and North America will decline from the 2020 level of above 1 g m⁻² a⁻¹ to lower than 0.3 g m⁻² a⁻¹ in 2050 and below 0.1 g m⁻² a⁻¹ in 2100 (Fig. S2d–f). Similarly, the emissions of OC (organic carbon) over the polluted regions will decrease from more than 0.2 g m⁻² a⁻¹ in 2020 to lower than 0.14 g m⁻² a⁻¹ and 0.1 g m⁻² a⁻¹ in 2050 and 2100, respectively (Fig. S2g–i). Accordingly, the annual mean aerosol optical depth (AOD) at 550 nm will decrease by 0.08-0.2 over the polluted regions including East Asia, South Asia, Africa, and South America simulated by CESM1, with the maximum reduction above 0.2 in 2050 and above 0.3 in 2100 (Fig. S3).

The regional mean changes in TCO and the emissions of aerosols and the precursors in 2050 and 2100 relative to the 2020 levels are further depicted in Fig. S4. The TCO levels over most of the subregions will be reduced in the future, with TCO values decreasing by 10–15% in 2050 and 15–25% in 2100 relative to the 2020 levels. The emission reductions of aerosols and their precursors are relatively strong with $SO_2$ emissions decreasing by more than 50% in 2050 and up to 90% in 2100 over most of the subregions relative to 2020, BC emissions decreasing by 30–70% in 2050 and 50–90% in 2100 and OC emissions decreasing by 10–40% in 2050 and 20–50% in 2100.

### Changes in temperatures and precipitations attributed to GHGs, aerosols, and tropospheric ozone under carbon neutrality
With the fully coupled Community Earth System Model version 1 (CESM1, see "Methods" section), this study investigates the individual and combined climate impacts of the key anthropogenic climate drivers including GHGs, aerosols, and tropospheric $O_3$ in the future of carbon neutrality. To assess the climate effects of anthropogenic emission changes on regional scale, the global land regions are divided into 21 subregions (Fig. S5). And CESM1 model shows good performance in capturing the spatial patterns and magnitudes of temperatures and precipitations for present-day climate on global and regional scales (Fig. S6). Figure 1 shows the spatial distributions of the simulated changes in annual mean surface air temperatures in GHG2050, AerGHG2050, ALL2050, and ALL2100 relative to the Baseline simulations (see "Methods" section), attributed to the strict control of anthropogenic emissions under carbon neutrality. Driven by the increased levels of GHGs, including $CO_2$, $N_2O$, and CFC-12 in 2050 relative to the 2020 values (see Table S1), surface air temperatures will change slightly over most of the globe, with maximum increases within 0.2 °C over Greenland (Fig. 1a). By further reducing the emissions of aerosols and precursors, surface temperatures are significantly

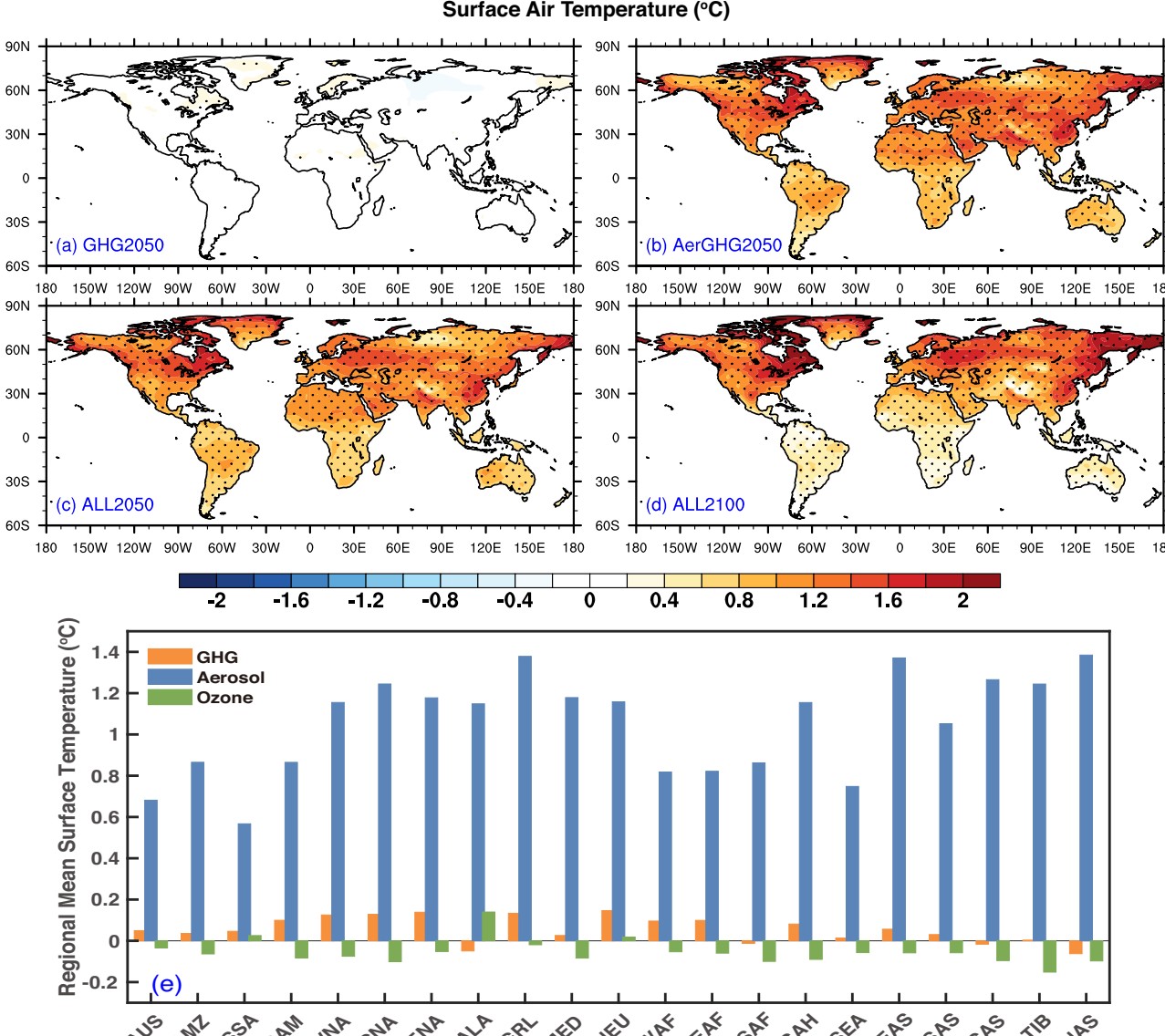

**Fig. 1 | Changes in annual mean surface air temperature.** Spatial distributions of changes in annual mean surface air temperature (°C) in GHG2050 (**a**), AerGHG2050 (**b**), ALL2050 (**c**), and ALL2100 (**d**), relative to Baseline (2020) and regional mean surface temperature changes in 2050 attributed to greenhouse gases (GHGs), aerosols, and tropospheric ozone (O₃) changes over 21 subregions (**e**) defined in Fig. S5. The stippled areas indicate statistical significance with 95% confidence from a two-tailed Student's *t*-test.

elevated in AerGHG2050, with maximum increases up to 2.0 °C over mid-to-high latitudes of the Northern Hemisphere relative to Baseline (Fig. 1b). Adding the reductions in tropospheric O₃, the warming will be slightly weakened in ALL2050 compared to AerGHG2050, relative to Baseline (Fig. 1c). The regional mean changes in surface air temperatures due to changes in GHGs, aerosol emissions and tropospheric O₃ levels in 2050 indicate that aerosol reduction-caused warming dominates climate change over all subregions with surface air temperatures increasing by 0.5–1.4 °C, noticeably higher than the GHG-caused surface temperature increases of below 0.2 °C (Fig. 1e). In contrast, the changes in tropospheric O₃ levels mainly induce cooling over most subregions, but the magnitudes of temperature changes are much weaker (<−0.2 °C) compared to the changes attributed to aerosols. By further controlling anthropogenic emissions by the end of the century, the air temperature increases are slightly enhanced in the ALL2100 simulations, especially in the mid-to-high latitudes but weakened over the tropics, compared to those in ALL2050 (Fig. 1d). It indicates that even negative CO₂ emissions after 2050 following the

SSP1-1.9[29] would not be enough to compensate for the excessive warming caused by further reductions of aerosols after 2050.

Note that the multi-model analyses based on SSP245, SSP245-GHG, and SSP245-Aer experiments under the medium emission pathway from DAMIP in CMIP6 suggest that the increases in the global surface air temperature in the mid of the century is primarily attributed to changes in GHGs rather than aerosols (Fig. S7). Specifically, relative to the 2020 level, surface air temperatures increase more than 1.2 °C over most of the land areas by the mid-of-century, with maximum above 1.4 °C in the north of 60 °N in SSP245 experiment (Fig. S7a), of which the changes in GHGs contribute the most with similar spatial pattern and magnitudes of temperature changes in SSP245-GHG (Fig. S7b). In contrast, changes in air temperatures driven by aerosols only are much weaker with magnitude <0.5 °C over most of the land areas in SSP245-Aer (Fig. S7c). The results are consistent with a recent study that GHGs contribute the most to projected changes in land surface air temperature under SSP2-4.5 emission pathway[30]. Furthermore, future precipitation changes are also dominated by GHGs rather

than anthropogenic aerosols as predicted under the SSP2-4.5 scenario[26]. Therefore, as opposed to the medium emission pathway, the unexpected amplified impacts of aerosols on climate changes should be taken seriously on the pathway to carbon neutrality.

Temperature changes are driven by perturbations of the earth's radiative balance, associated with reduced anthropogenic emissions[31]. The changes in effective radiative forcing (ERF) at top of the atmosphere (TOA) and at the surface in GHG2050, AerGHG2050, ALL2050, and ALL2100 compared to Baseline are illustrated in Figs. S8 and S9. Compared to Baseline, the changes in ERF at TOA due to changes in GHG concentrations are weakly positive, with a maximum of 0.5-1.0 W/m² over North America, South America, and West Africa (Fig. S8a). The TOA ERF changes between AerGHG2050 and Baseline are much larger, reaching 4.5 W/m² over East Asia and 2.5 W/m² over Northern Europe, North America, and East Africa (Fig. S8b). The positive TOA ERF anomalies associated with changes in GHGs and aerosols in 2050 indicate that more solar radiations will be captured by the earth system. However, negative TOA ERF anomalies up to −3.0 W/m² are witnessed over Greenland, North Africa, the Middle East, and Central Asia in AerGHG2050 compared to Baseline, where the radiative impacts of absorbing aerosols (e.g., BC in this study) are enhanced by the bright surface with a high albedo[32]. The decreases in these absorbing aerosols lead to a negative ERF anomaly at TOA. The ERF changes in ALL2050 compared to Baseline are of similar spatial patterns and magnitudes to those in AerGHG2050, suggesting that the decreases in O₃ have relatively weaker climate impacts than aerosols in 2050 (Fig. S8c). The regional mean ERF changes at TOA in 2050 due to changes in GHG concentrations, aerosol emissions, and tropospheric O₃ suggest that aerosol-caused ERF changes (within ±2.0 W/m²) overtake the GHG and tropospheric O₃ induced ERF changes (within ±0.5 W/m²; Fig. S8e). The ERF changes at TOA are further enhanced by the end of the century in ALL2100 (Fig. S8d), primarily driven by the continuing decreases in aerosol emissions (Figs. S1–S3). Note that the spatial distributions of ERF changes at the surface in GHG2050, AerGHG2050, ALL2050, and ALL2100 relative to Baseline are similar to the ERF changes at TOA (Fig. S9). The reduction in aerosol emissions contributes the most to the increases in ERF at the surface while changes in GHG concentrations and tropospheric O₃ play a less significant role. The changes in ERF at TOA and surface support the temperature changes that the reductions in aerosols and their precursor emissions contribute dominantly to future warming under the carbon neutrality scenario.

The spatial distributions of annual mean precipitation changes caused by anthropogenic emission changes compared to Baseline are shown in Fig. 2. Precipitation anomalies caused by changes in GHGs are primarily identified over the tropical oceans, with significant positive anomalies over the Western Pacific and insignificant negative anomalies over the Eastern Pacific and Indian Ocean (Fig. 2a). The enhanced rainfall over the tropical Western Pacific agrees with previous findings[33]. By further reducing aerosol precursors, precipitation changes are amplified globally in AerGHG2050 compared to Baseline, with increased precipitation over the Northern Hemisphere, especially along the tropics, and decreased precipitation over many oceans of the Southern Hemisphere (Fig. 2b), suggesting a northward shift of the intertropical convergence zone due to aerosol reductions that largely occur over the Northern Hemisphere[34]. The spatial pattern of precipitation changes in ALL2050 is similar to that in AerGHG2050 (Fig. 2c), indicating that the reductions in tropospheric O₃ levels have little impact on mean precipitations. The regional attributions of mean precipitation changes show that the decreases in aerosols generally increase precipitation over the global land regions, with maximum precipitation increases of up to 0.3 mm/day over Southeast Asia, East Asia, and South Asia. In contrast, changes in GHGs and tropospheric O₃ levels generally have much weaker impacts on precipitations, with magnitudes within ±0.05 mm /day (Fig. 2e). In ALL2100, the spatial

pattern of precipitation changes is similar to that of ALL2050, but with relatively larger amplitudes, consistent with the much lower aerosol levels by 2100 (Fig. 2d). The crucial impacts of aerosols on precipitations are consistent with previous works that higher aerosol concentrations reduce cloud-droplet size and prolong cloud lifetime, leading to reduced rainfall[35,36].

The aerosol reduction-induced warming can also lead to more evaporation and ultimately result in more precipitation[37]. Thus, the changes in precipitation rates caused by anthropogenic emission changes under carbon neutrality are also closely related to the increase in water vapor with warming (Fig. S10). Compared to Baseline, except with weak increases in regions north to 60°N, the specific humidity changes little over other regions of the globe in GHG2050 simulations (Fig. S10a). But in AerGHG2050, specific humidity is much amplified, with large positive anomalies along tropical regions up to 1 g/kg compared with Baseline (Fig. S10b), which may explain the increased precipitations there (Fig. 2b). The magnitude and spatial pattern of humidity changes in ALL2050 relative to Baseline are similar to those in AerGHG2050, suggesting that changes in tropospheric O₃ levels contribute little to the changes in humidity and thus precipitations (Fig. S10c), in accordance with their small role in modulating temperature changes. With relatively enhanced warming in ALL2100 than ALL2050 relative to Baseline (Fig. 1), specific humidity changes are amplified in 2100 under the control policy of carbon neutrality. Overall, the decrease in aerosols and their precursors play a dominant role in modulating the changes in not only mean air temperatures but also precipitations.

It is worth noting that the spatial pattern of changes in surface air temperature (Fig. 1c) and precipitation (Fig. 2c) in ALL2050 simulated by CESM1 are similar to those projected by the multi-model ensemble mean of 13 global climate models in CMIP6 (Table S2) under SSP1-1.9 scenario (Fig. S11) with statistically significant spatial correlations above 0.8 for temperature change and 0.4 for precipitation change. Specifically, large increases in air temperature locate over mid-to-high latitudes of the Northern Hemisphere and increased rainfall anomalies mainly over the north of the equator, implying the crucial role of changing anthropogenic emissions in the future climate. It should be noted that the magnitudes of changes in air temperatures and precipitations in CESM1 are higher than those in multi-model results under SSP1-1.9. To be more specific, the global mean changes in air temperatures and precipitation in 2050 relative to 2020 simulated by CESM1 are 0.85 °C and 0.07 mm/day, respectively, which are higher than the changes predicted by CMIP6 averages (0.60 ± 0.50 °C and 0.04 ± 0.24 mm/day) related to the larger ERFs due to aerosol reductions and thus stronger climate sensitivities. Moreover, the global mean future changes in surface air temperatures and precipitations in 2100 relative to 2020 levels are 0.92 °C and 0.10 mm/day in CESM1, also exceeding the CMIP6 averages (0.40 ± 0.60 °C and 0.04 ± 0.24 mm/day).

## Changes in extreme weather events attributed to GHGs, aerosols, and ozone under carbon neutrality

Extreme temperatures and precipitations are more influential to humans and societies. In this section, we focus on changes in extreme events including (dry) heat waves, humid heat waves, and extreme precipitation indices (see "Methods" section). In GHG2050 (Fig. 3a, e, i), the average frequency, length, and amplitude of extreme events are all quite moderate with average heat wave frequency (HWF, days/year), duration (HWD, days/event) and amplitude (HWA, °Cday) lower than 5 days/year, 4 days/event, and 0.25 °C/day, respectively, over most of the globe. When combined with aerosol emission abatement, heat wave frequency increases dramatically, along with increased mean duration and magnitude in AerGHG2050 (Figs. 3b, f, j), with HWF, HWD, and HWA increasing to more than 40 days/year, 20 days/event and 0.75 °C/day globally. The spatial patterns and magnitudes of

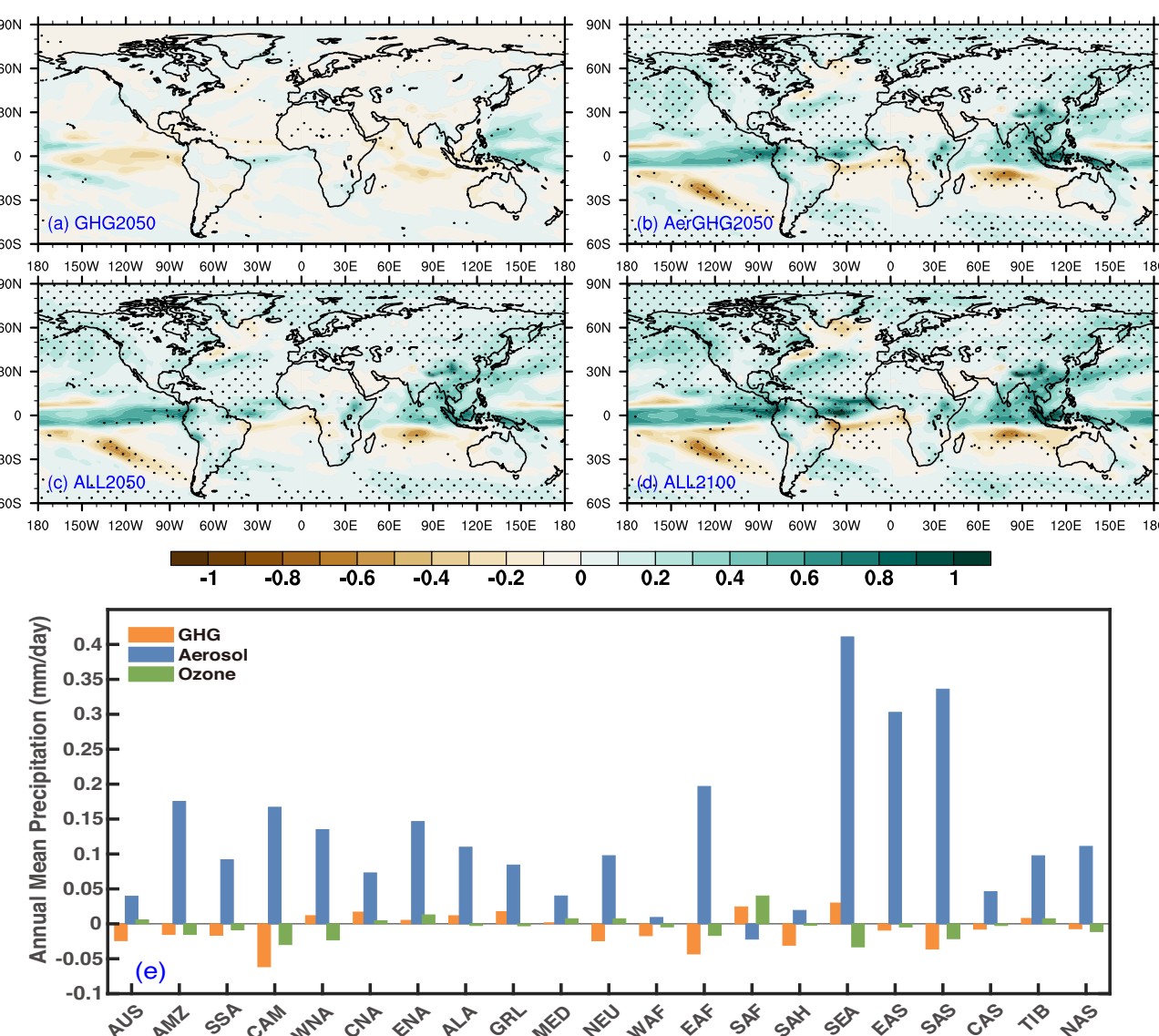

**Fig. 2 | Changes in annual mean precipitation.** Spatial distributions of changes in annual mean precipitation (mm/day) in GHG2050 (**a**), AerGHG2050 (**b**), ALL2050 (**c**), and ALL2100 (**d**), relative to Baseline (2020) and regional mean surface temperature changes in 2050 attributed to greenhouse gases (GHGs), aerosols, and tropospheric ozone ($O_3$) changes over 21 subregions (**e**) defined in Fig. S5.

heat wave indices in ALL2050 are similar to those in AerGHG2050 (Fig. 3e, g, k), but tropospheric $O_3$ abatement will have a small dampening effect on the occurrence of heat wave events (Fig. S12a). Further decreases in anthropogenic emissions by 2100 favor more, longer-duration, and stronger heatwaves such. Mean HWF and HWD will exceed 50 days/year and 28 days/event globally (Fig. 3d, h) and HWA increases to 1.5 °C/day over mid-to-high latitudes (Fig. 3l), implying that the aerosol-dominated warming effect adds to excessive heat wave events by the end of the century (Fig. 3d). Therefore, substantial reductions in $O_3$ precursors and GHGs emissions would have to be implemented to counteract the harmful climate consequences of future aerosol decline, which is also highlighted in SSP1-1.9 that $CO_2$ emissions would decline to net zero around 2050, followed by net negative $CO_2$ emissions[29].

Reductions in aerosols have a similar amplification effect on humid heat waves as dry heat waves. Reductions in aerosol emissions favor increased humid heat waves with longer duration, and stronger amplitudes (Fig. S13). Specifically, the humid-HWF, humid-HWD,

and humid-HWA (see "Methods" section) increase from less than 4 days/year, 4 days/event, and 0.25 °C/day in GHG2050 (Fig. S13a, e, i) to more than 30 days/ year, 24 days/event, and 0.5 °C/day over most of the globe in AerGHG2050 (Fig. S13b, f, j), respectively. The enhancement of aerosol emission changes on the occurrences of humid heat waves outweighs the moderate effect of $O_3$ and GHGs in 2050 (Fig. S12b). As aerosol emissions are further reduced by 2100, both dry and humid heat waves will threaten larger populations that increase the need to advance climate change adaptations.

We further evaluate the impacts of anthropogenic emissions on extreme precipitation indices (see "Methods" section). Compared with Baseline, Pretotal, the annual total precipitations on wet days (rainfall rate (RR) ≥1 mm/day), are reduced over many land regions in GHG2050 with negative anomalies of more than −20 mm dominating the tropics, Northern Europe and North Asia (Fig. 4a), whereas further reductions in aerosols largely increase Pretotal with significant positive anomalies higher than 40 mm over most of the globe especially the mid-to-high latitudes of Northern Hemisphere in AerGHG2050

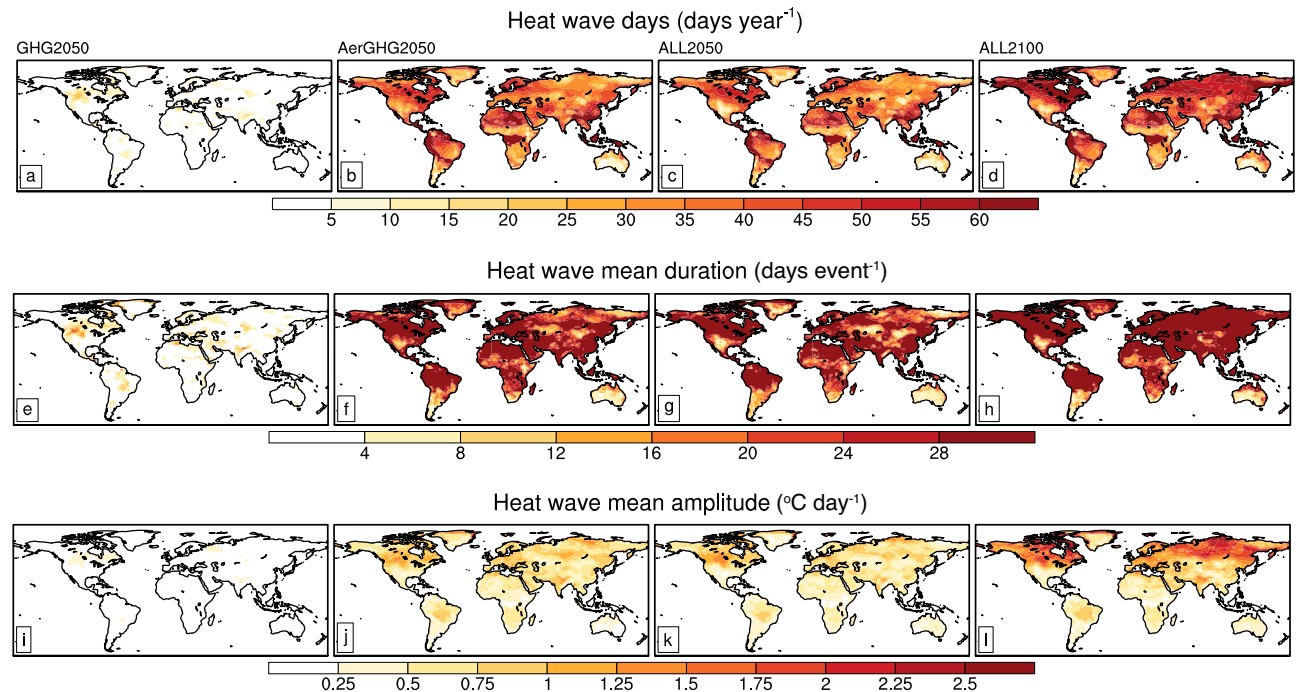

**Fig. 3 | Global maps of heat wave indices.** Heat wave days (days/year), mean duration (days/event) and amplitude (°C/day) in GHG2050 (**a, e, i**), AerGHG2050 (**b, f, j**), ALL2050 (**c, g, k**), and ALL2100 (**d, h, l**) simulations.

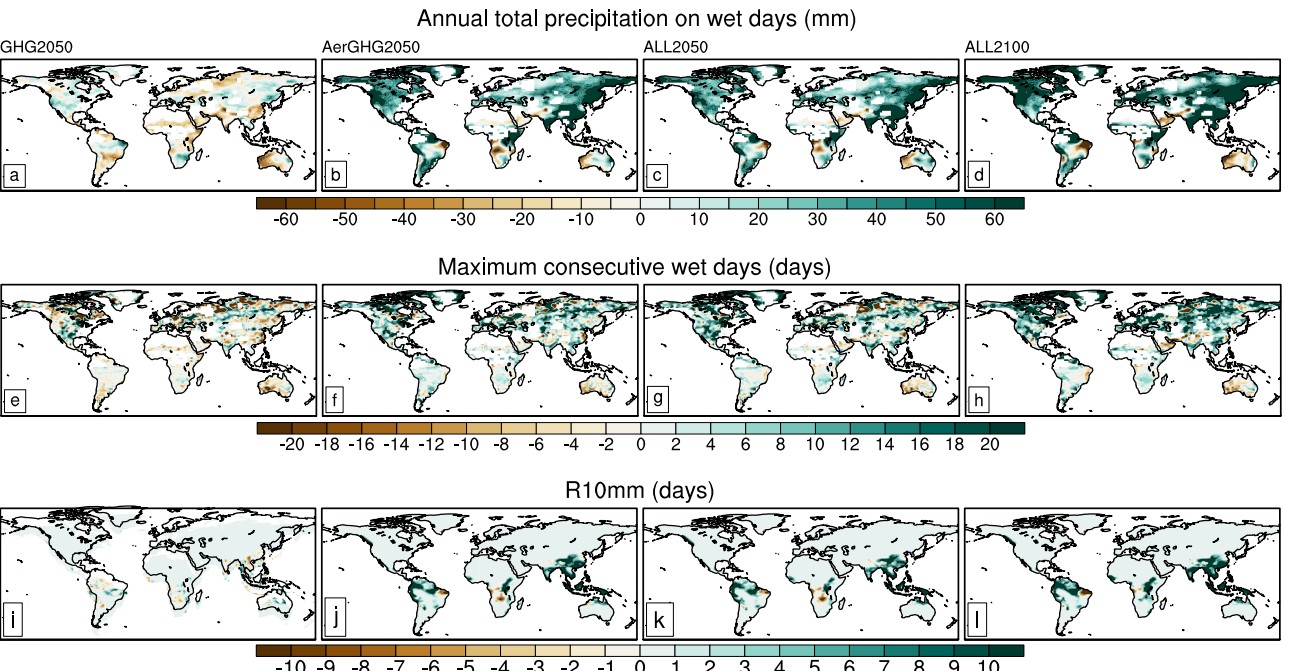

**Fig. 4 | Global maps of extreme precipitation indices.** Changes in annual total precipitation on wet days (mm), maximum consecutive days (days) and heavy precipitation days (R10mm, days) in GHG2050 (**a, e, i**), AerGHG2050 (**b, f, j**), ALL2050 (**c, g, k**) and ALL2100 (**d, h, l**), relative to Baseline (2020).

(Fig. 4b). Tropospheric $O_3$ levels have relatively small impacts on Pretotal (Fig. 4c). The regional attributions of Pretotal indicate reductions in aerosols have positive contributions to Pretotal while changes in GHGs and tropospheric $O_3$ have weak negative contributions in 2050 (Fig. S12c). The impacts of changes in GHGs, aerosols and precursor emissions, and tropospheric $O_3$ levels on total wet days are similar to those of Pretotal (Fig. 4e–h). Continued strict controls of

anthropogenic emissions by 2100 lead to more Pretotal and wet days, mainly contributed by the much lower aerosols by 2100 (Fig. 4d, h). Note that the enhancements of reductions in anthropogenic emissions on R10mm, total days of moderate precipitation with RR≥10 mm/day are mainly located over tropical regions, of which the reduced aerosols contribute to increased R10mm while changes in GHGs and $O_3$ have the opposite effects (Fig. 4i–l).

## Discussion

In this study, we investigate the climate impacts of anthropogenic emission changes under the global carbon neutrality scenario using the fully coupled climate model CESM1. The distinct climate effects of changes in GHGs, aerosol emissions, and tropospheric $O_3$ levels in the future are assessed individually. While DAMIP simulations have been conducted to attribute the observed climate change to individual effects of various anthropogenic forcings and to project their effects under the SSP2-4.5 scenario, here we focus on the carbon neutrality pathway following the SSP1-1.9 scenario with multiple CESM1 long-term simulations. We show that the impacts of aerosol reductions on climate and extreme weather events far outweigh those of GHGs and tropospheric $O_3$ under carbon neutrality, which reverses the knowledge that the changing GHGs dominate the future climate changes as predicted in the middle of the road pathway. The anthropogenic forcers including GHGs and aerosols are the main drivers of the past and future changes in the climate as well as the extremes, as extremes are parts of the climate system. In particular, in the future of carbon neutrality, reduced aerosols enhance the warming effect caused by GHGs. Changes in extremes at the global and regional scales are a direct consequence of the enhanced radiative forcing, and the associated global warming and/or its resultant increase in the water-holding capacity of the atmosphere based on the Clausius-Clapeyron relationship through thermodynamical processes[29]. At global scale, the number of heatwave days and the length of heatwave seasons in many regions scale well with global mean temperatures[38,39], and changes in annual maximum one-day precipitation are proportional to global mean temperature changes[40,41]. At the local and regional scales, aerosol concentrations changes are also a key factor in controlling and modulating temperatures and extremes[42], through the combined effects of the atmospheric energy balance, dynamical adjustment, and vertical structure of forcing[43,44].

This study is subject to several limitations and uncertainties. Simulating the climate forcing of changes in $CO_2$ concentration, a long-lived climate forcer, generally requires a long spin-up for the model to reach equilibrium[45]. Thus, the equilibrium climate response to GHG changes is likely underestimated in our simulations. The transient and equilibrium responses of both surface air temperature and precipitation to $CO_2$ increases are broadly similar in spatial patterns over most of the globe among different forcing scenarios in CESM1 simulations[46]. If the transient simulations are performed, the patterns of temperature/precipitation changes in response to GHGs changes would be similar to those obtained through the current equilibrium simulations following SSP1-1.9, but the responses to GHGs changes would be even weaker in transient simulations as compared to equilibrium simulations[47], which would not alter the finding in this study that the climate effects of aerosol reductions far outweigh those of GHGs under carbon neutrality. However, as previous studies have pointed out the remarkable differences in climate response between transient and stabilized scenarios[47,48], further in-depth exploration is needed to explore the differences and better understand the future projections of climate change.

Moreover, large model uncertainties exist in projections of climate response to anthropogenic forcing, and CESM1 is relatively more sensitive to anthropogenic forcings. The equilibrium climate sensitivity in CESM1-CAM5 is 4.1 °C[49], exceeding the upper tail of the very likely range of 2.5– 4 °C obtained by multi-model simulations in CMIP6[29]. CESM1 generally has a higher ERF due to aerosol-cloud interactions than other climate models[50], partly explaining the relatively stronger responses in temperature and precipitation to the anthropogenic forcings in ALL2050 than the ensemble means of multi-models in CMIP6 by 2050 (Fig. S11).

The interactions of anthropogenic aerosols with radiation and clouds are the largest source of uncertainty in the estimation of effective radiative forcing[51]. Aerosol ERF at TOA in present-day (2000) relative to the preindustrial (1860) level simulated by CESM1 is $-1.37$ W/m²[50] and $-1.44$ W/m² with an additional 5% applied to account for land-surface cooling[52], relative higher than the multi-model averages of $-1.23 \pm 0.48$ W/m² in the Coupled Model Intercomparison Project Phase 5 (CMIP5) over 1860–2000 and $-1.11 \pm 0.38$ W/m² in the CMIP6 over 1850–2014[29]. Here, the global mean changes in ERFs driven by aerosol reductions in 2050 relatively to 2020 in the nine global climate models used in Zelinka et al.[50] are estimated and given in Fig. S14. Though the multi-model average of aerosol-driven changes in ERF is lower than that in CESM1, it is still far beyond the global mean changes in ERF due to GHGs changes (0.01 W/m²), as shown in Figure S8a. Therefore, the main finding that aerosol reductions dominate climate changes toward carbon neutrality target would not be changed with multi-model simulations. Still, it is of great significance for the community to conduct multi-model intercomparison of future climate change in response to individual anthropogenic forcings under the carbon neutrality scenario.

On the other hand, absorbing and scattering aerosols have opposite effects on climate. For example, the warming induced by decreases in scattering aerosols could be partly offset by the net cooling from reduced absorbing aerosols under the carbon neutrality scenario[53]. Their relative roles in future climate toward carbon neutrality warrant further investigations. Moreover, the future global mean concentrations of GHGs levels and tropospheric $O_3$ were prescribed following SSP1-1.9 without considering the interactions between aerosols and the tropospheric gas-phase species. Aerosols influence $O_3$ chemistry by enabling heterogeneous and multiphase reactions[54]. For example, decreased aerosols in China may increase $O_3$ by slowing down the sink of hydroperoxy radicals, thus speeding up ozone production[55]. Further, heterogeneous nitrate radical ($NO_3$) and dinitrogen pentoxide ($N_2O_5$) reactions on sulfate aerosol particles might reduce the global tropospheric average $O_3$ burden by around 9%[56]. Thus, the potential impacts of the interactions among air pollutants on climate mitigation deserve further efforts. Moreover, as discussed above, climate change can modulate air pollutants by affecting their accumulations or productions, which in turn can feedback on climate, but such interactions are neglected in this work. Therefore, further detailed studies are needed to address the above modeling issues.

It should be noted that the warming due to aerosol reductions toward carbon neutrality are actually a committed warming due to rising long-lived GHGs, and the aerosol reductions simply unmask that GHGs-induced warming. In addition, despite of negative climate impacts of future aerosols emission reductions, the benefits of carbon neutrality on air quality and global health burden are worthy of attention[17,57].

## Methods

### Model description

In this study, the climate effects of anthropogenic emission changes under carbon neutrality are investigated through model simulations with the fully coupled Community Earth System Model version 1 (CESM1)[63]. The atmosphere component uses the Community Atmosphere Model version 5 (CAM5) with a horizontal resolution of 1.9° x 2.5° and 30 vertical layers. Major aerosol species, including BC, primary organic matter (POM), secondary organic aerosol (SOA), sulfate (SUL), mineral dust, and sea salt, are predicted in four lognormal size distribution modes (i.e., Aitken, accumulation, coarse and primary carbon modes)[58]. Aerosols are internally mixed within each mode. The land component is the Community Land Model version 4 (CLM4) and the ocean component employs the Parallel Ocean Program version 2 (POP2). CESM1 incorporates both aerosol direct and indirect effects including the cloud albedo and cloud lifetime effects[59]. The CAM5 in CESM1 features improved aerosols representation and includes aerosol–cloud interactions, demonstrating improved

performance relative to CAM4[60]. Additionally, a few aerosol-related schemes in CAM5, such as aerosol convective transport and wet deposition, are modified to enhance the model performance[61]. Aerosols are well simulated in North America and Europe but somewhat underestimated in East Asia and remote regions[20]. The three-dimensional $O_3$ concentrations are prescribed to estimate the radiative effect of $O_3$.

CESM1 has been widely used to investigate the local and global climate effects of emission and concentration changes in GHGs and air pollutants[21,62]. The model realistically simulates the observed warming over the 20th-century[63]. In this study, the SSP1-1.9 scenario in CMIP6 is used to represent the carbon neutrality pathway, which is a sustainable development pathway yielding an approximate anthropogenic radiative forcing of 1.9 W m$^{-2}$ in 2100 relative to the preindustrial level. The SSP1-1.9 scenario is developed to meet the 1.5 °C global warming target set by the Paris Agreement[64-66] and has been widely used as the carbon neutrality scenario to examine the future air quality and climate in many studies[67,68].

## Experimental design
Five sets of climate simulations (Table S3) are performed to investigate the climate effects of changes in anthropogenic emissions under carbon neutrality, with perturbed GHGs concentrations, aerosols and precursor emissions, and tropospheric $O_3$ levels following SSP1-1.9. Although for a long time, many numerical studies investigate the climate under the 1.5 °C/2 °C global warming as a transient phase, the 1.5 °C/2 °C global warming limits are set by the Paris agreement with quasi-stabilized scenarios[47]. Recent studies have revealed substantial differences in climate responses between transient and stabilized climates and suggested that it would be valuable to design experiments in meeting the needs of decision-makers given that the Paris Agreement is implicitly targeting stabilized global warming levels[48,69]. Our study aims to understand the future climate effects of different anthropogenic forcings under quasi-equilibrium conditions for the development of climate change adaptation policies.

We focus on climate changes over the mid (2050) and end (2100) of the 21st century, compared with the present-day climatology (2020). In the Baseline simulation, global mean GHG concentrations, aerosols, and their precursor emissions, and tropospheric $O_3$ levels are prescribed from the CMIP6 data and all fixed at the 2020 levels under the SSP1-1.9 scenario. Specifically, the global mean concentrations of major GHGs, including $CO_2$, $CH_4$, $N_2O$, and halocompounds (CFC-11 and CFC-12) for the 2020 level are fixed at 414 ppm, 1884 ppb, 331 ppb, 218 ppt, and 495 ppt in the Baseline simulations (Table S1). Emissions of aerosols and their precursors in the model simulations follow the emission pathway of SSP1-1.9. The $O_3$ concentrations in 2020, 2050, and 2100 are the ensemble averages of six global climate models (GCMs) simulations of SSP1-1.9 from the Scenario Model Intercomparison Project (ScenarioMIP) in CMIP6 (Table S4). Although the $O_3$ concentrations are prescribed throughout the atmosphere, the climate effects of $O_3$ are dominated by the changes in tropospheric $O_3$[1]. The ScenarioMIP GCMs show good performance in simulating historical tropospheric $O_3$[70,71].

In GHG2050, the GHG concentrations are held at the future 2050 levels under SSP1-1.9, along with aerosols and precursor emissions and tropospheric $O_3$ fixed at 2020 levels. Comparing GHG2050 and Baseline isolates the quasi-equilibrium climate effects of GHG changes by 2050 under carbon neutrality. In AerGHG2050, both GHG concentrations and aerosol emissions are held at the future 2050 levels under SSP1-1.9, along with tropospheric $O_3$ set at the current 2020 level. Thus, comparing AerGHG2050 and GHG2050 isolates the climate effects of aerosol changes by 2050 under carbon neutrality. In the ALL2050 simulations, GHG concentrations, aerosol emissions, and tropospheric $O_3$ levels are all set at the 2050 levels under SSP1-1.9.

Comparing ALL2050 and AerGHG2050 represents the climate effects of tropospheric $O_3$ changes by 2050 under SSP1-1.9. We also perform the ALL2100 simulations in which GHGs concentrations, aerosol emissions and tropospheric $O_3$ are all set to the 2100 levels under SSP1-1.9, which allows comparison with Baseline to estimate the combined climate impacts of carbon neutrality by the end of the 21st century. For each of the experiments, 3 ensemble members are performed with a small initial perturbation to atmospheric temperature. All model simulations are run for at least 200 years of which the last 100 years are used in the analysis of climate impacts.

## DAMIP multi-model simulations
In this study, we also place a special focus on the future global surface air temperature changes under a medium emission pathway and the differences of anthropogenic effects between the medium emission pathway and those under the carbon neutrality. As mentioned above, the DAMIP simulations of aerosols-only, stratospheric-ozone-only, GHG-only, solar-only, and volcanic-only forcing are carried out as part of the CMIP6 to facilitate an improved estimation of the global and regional climate response to individual forcing. And all the future experiments in DAMIP are conducted under the medium emission scenario SSP2-4.5.

Here, we conduct the multi-model analyses of all-forcing simulations under SSP2-4.5 scenario (SSP245), simulations of GHG-only (SSP245-GHG) and aerosol-only forcing (SSP245-Aer) from the DAMIP experiments to quantify the individual contributions of GHGs and aerosols to future surface air temperature changes in the mid-of-the-century under the medium emission pathway (Table S5) and compare those with the anthropogenic effects under carbon neutrality.

## Climate extreme indices
To measure the impacts of anthropogenic changes on extreme weather events, five extreme indices derived from daily maximum temperatures (Tmax) and daily rainfall rate (RR) are analyzed in this work: heat waves, humid heat waves, heavy precipitation days (R10mm), maximum length of wet days (CWD), and total precipitation of wet days (Pretotal). Wet days here mean daily precipitation above 1 mm/day (RR≥1 mm/day). Different from (dry) heat waves identified as extreme hot days[72,73], humid-heat waves are compound hazards of consecutive hot and humid days, defined by daily maximum wet-bulb temperatures (TWs), an integral measure of humidity and air temperatures. Using the algorithm described in Davies-Jones[74], the daily maximum TW is calculated from the daily maximum air temperature, daily mean relative humidity, and daily mean surface pressure. The co-occurrences of humid and hot days would amplify the negative impacts of excessive heat on human and mortality rates[75-77]. Detailed definitions of the five extreme indices are provided in Table S6. Among the five extreme indices, heat waves and humid heat waves consist of several days of extreme high temperatures/wet-bulb temperatures, which entail several aspects of interest including heat wave frequency (HWF, days/year), mean duration (HWD, days/event) and amplitude (HWA, °C/day). Frequency means the annual total number of days of (humid) heat waves, duration means the average number of consecutive days during (humid) heat waves, and amplitude means the average exceedance of daily Tmax/TW relative to the threshold of (humid) heat waves[78]. Note that heat waves and humid heat waves are identified based on the present-day thresholds derived from daily Tmax/TW in the Baseline simulations (see Table S6).

In this work, the present-day climatology of air temperatures and precipitation is evaluated against the ERA5 reanalysis at 0.25° × 0.25° resolution on 2020. ERA5 is the latest fifth-generation reanalysis global atmosphere dataset from the European Centre for Medium-Range Weather Forecasts (ECMWF)[79] (see Fig. S6).

## Data availability

The meteorological fields for 2020 level can be obtained from ERA5 reanalysis data (https://www.ecmwf.int/en/forecasts/datasets/reanalysis-datasets/era5). The multi-model outputs from ScenarioMIP and DAMIP in CMIP6 can be found at https://esgf-node.llnl.gov/search/cmip6/. The processed modeling data are available at https://doi.org/10.5281/zenodo.10013045.

## Code availability

The codes that support the findings of this study are fully available from the corresponding author (yang.yang@nuist.edu.cn) upon request.

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

## Acknowledgements

This research was supported by the National Natural Science Foundation of China (grant 42293323, H.L., 42105166, P.W., and 41975159, Y.Y.), the National Key Research and Development Program of China (grant

2020YFA0607803, Y.Y., and 2019YFA0606800, H.L.), Jiangsu Science Fund for Distinguished Young Scholars (grant BK20211541, Y.Y.), and the Jiangsu Science Fund for Carbon Neutrality (grant BK20220031, H.L.). L. Ruby Leung acknowledges the support by the U.S. Department of Energy (DOE), Office of Science, Office of Biological and Environmental Research (BER), as part of the Earth and Environmental System Modeling program. The Pacific Northwest National Laboratory (PNNL) is operated for DOE by Battelle Memorial Institute under contract DE- AC05-76RLO1830.

## Author contributions

Y.Y. conceived the research and directed the analysis. P.W. conducted the model simulations, performed the analysis, and drafted the manuscript. D.X. helped to organize and revise the manuscript. L.R., J.T., L.L., and H.L. discussed the results and helped to improve the manuscript.

## Competing interests

The authors declare no competing interests.
