## [Peer Review File · Nature Communications]

Aerosols overtake greenhouse gases causing a warmer climate and more weather extremes toward carbon neutralityREVIEWER COMMENTS

Reviewer #1 (Remarks to the Author):

The authors present the impacts of greenhouse gases, aerosols, and tropospheric O₃ on climate and extreme weather events under a carbon neutrality pathway. With the Community Earth System Model version 1 (CESM1), the authors find that the climate effects of aerosol reductions far outweigh those of GHGs and tropospheric O₃ under carbon neutrality, which reverses the knowledge that the changing GHGs dominate the future climate changes as predicted in the medium forcing pathway. This study addresses very important questions that how the climate and extreme weather events evolve under a carbon neutrality scenario and which external forcing contributes the most to the climate change. The paper presents novel ideas and analysis and the scientific methods and assumptions are valid and clearly outlined. Before this paper can be considered for publication, I have a few minor comments need to be addressed.

1. It is interesting that the carbon neutrality scenario (SSP1-1.9) presents different contributions of aerosols on climate as compared to SSP2-4.5. The author can compare aerosol reductions under SSP1-1.9 with those under SSP2-4.5.
2. According to the model simulations, GHGs have very slight impacts on temperatures and precipitations compared to aerosols toward carbon neutrality. It is the most significant finding and reverses the current climate projection in other scenarios. Does each of the 3 model ensemble members exhibit the same result?
3. I wonder why you selected to run the simulations using fixed forcing for 2020, 2050, and 2100 to get a quasi-equilibrium response. Some studies presented climate responses using the transient simulations. It is not necessary that the author should perform new simulation, but perhaps you should discuss the possible difference between transient and equilibrium responses.
4. Recently, Wang et al. (2023, IJOC) have analyzed influences of anthropogenic and natural forcings on future changes in precipitation projected by the CMIP6-DAMIP model. It would be better to cite this paper to enhance the result obtained from this manuscript (Wang, Y., T. Zhao*, L. Hua, X. Guan, C. Xu, and X. Chen, 2023: *Int. J. Climatol.*, doi: 10.1002/ioc.8064)
5. About the climate Extreme Indices, the humid heat waves are defined based on daily maximum wet-bulb temperature (TW). The question is how TW is calculated? Is it an output of the model?
6. Orders of figures needed to be adjusted. For example, the supporting figure starts from Figure S3.
7. L115: 154: Table S3 to Table S2
8. L170-178: Are there any published works to support the finding here?

9. L318: SSP119 to SSP1-1.9

10. Missing y-axis labels in Fig. S6d.

11. Inconsistent fonts, e.g., Fig. S3.

Reviewer #2 (Remarks to the Author):

This study assesses individually the global impacts of changing GHGs, aerosols, and tropospheric O₃ levels following a carbon neutrality pathway on climate and extreme weather events by using the Community Earth System Model version 1 (CESM1). The author found that the global changes in climate and extremes by 2050 and 2100 will be considerably stronger due to the emission reductions in aerosols and precursors than those due to GHGs and tropospheric O₃ changes. The study suggests that substantial reductions in GHGs and tropospheric O₃ are necessary to reach the 1.5 °C warming target.

The finding is interesting although expected as the CESM1 model has a strong (negative) aerosol radiative forcing at present-day. A reduction in aerosol emissions in the future (e.g., 2050) implies a strong warming. As only one model (which may have too strong aerosol radiative forcing) is used in this study, the authors may consider the inclusion of results from the other models. The uncertainty of aerosol radiative forcing in the CESM1 model and implications to the results of this study should be discussed.

The authors mentioned 12 global climate models in CMIP6 under SSP1-1.9 scenario and used the multi-model ensemble mean (Figure S11 and Table S5). It would be interesting to include these model results in the analyses.

This study used the approach for the equilibrium climate state (time slides) rather than the transient climate change approach commonly used in the IPCC AR6. This may make the comparison with the DAMIP multi-model simulations (SSP245) ambiguously.

Although the impacts of aerosol on temperature and precipitation are well studied, a discussion on the mechanisms of aerosol impacts on extreme weather events may be required to be included as this is the most interesting part of this study.

The writing of this paper needs to be improved:

1. The abstract is wordy and should be much condensed.
2. The appearance of the first figure is Fig. S3 (line 123). This seems not in the right order of figures, e.g.,

the first figure appearing in the text should be Figure S1.

3. When reading the results sections, it is unclear that these results were obtained using the CESM1 model. CESM1 is mentioned in the abstract and then in the Method section. E.g., for the results in Lines 121-125, how and which model is used to derive them?

4. Line 257, what are “HWF, HWD, and HWA”. They need to be defined.

5. Line 281, it uses “Pretotal” while “Pretol” is used in Figure 4 caption.

Reviewer #3 (Remarks to the Author):

Review of “Aerosols overtake greenhouse gases causing a warmer climate and more frequent extreme weather events toward carbon neutrality” by Pinya Wang, Yang Yang, Lili Ren, Daokai Xue, Jianping Tang, L. Ruby Leung, Hong Liao

The study investigates the relative contribution on future climate from the different climate forcings: well-mixed GHGs, aerosols and precursors, and ozone following a carbon-neutrality scenario using the model CESM. The main results are not ground-breaking or surprising, given the wealth of research showing the future changes in climate due to strong aerosol reductions (many cited in the manuscript). In my opinion the main value of the study is that it demonstrates that under a carbon-neutrality scenario, aerosols largely dominate the future climate response, something that is still perhaps overlooked by the community and carries an important political message. The study has a well-defined scientific question, the methodology followed is clearly described and the manuscript is well written.

General comments:

- It seems like the CESM simulations show quite stronger responses in temperature and precipitation than the mean of 12 CMIP6 models under scenario SSP1-1.9 by 2050 (Figs 1 and 2 vs. Fig S11). If scenario SSP1-1.9 by definition limits warming to around 1.5 degrees based on a multi-model ensemble (See table SPM1 in IPCC, 2021: SPM), where does CESM stand in this ensemble? Is CESM more sensitive to aerosol reductions than the multi-model mean? I think there should be enough literature comparing CESM with other models to try and better understand where the model stands in terms of aerosol responses. Would the main conclusions be the same if the model was very insensitive to aerosol forcing? Or are these differences related to the semi-equilibrium response as opposed to the transient one?

- The authors mention many and very valid limitations and uncertainties, but miss the most important one, which is model uncertainty. The qualitative comparison between Figs. 1-2 and Fig S11 seems insufficient to claim that CESM represents the real climate response to the studied forcings. This is not a particular argument against the model, the same could be said about any model. I do think that the authors need to clearly state this in the discussion and wherever relevant in the manuscript. Furthermore, this relative weakness could be turned into a powerful message: encourage the

community to pursue in future multi-model detection and attribution assessments this type of best case, low carbon scenarios which may still carry important negative consequences.

- In addition, despite of apparent negative climate impacts of future aerosols emission reductions, major health benefits could be expected in regions with air pollution, something that in my opinion deserves to be highlighted in the manuscript. In fact, the warming due to aerosol reductions is actually a committed warming due to rising long-lived GHGs, the aerosols are simply masking that warming. Acknowledging the benefits of anthropogenic aerosols reductions could help frame the message in a more balanced manner.

- The fact that the climate response is stronger by 2100 than by 2050 means that at least according to CESM, not even negative carbon dioxide emissions after 2050 following the SSP1-1.9 would be enough to compensate for the further reductions of aerosols after 2050? If so, this can be an important message to convey.

Specific comments:

-Lines 79-81: Is this sentence necessary? It seems a bit isolated from the context. Why Portugal?

-Line 146: Missing y-axis label in Fig S6d.

-Line 188: Seems like South America has a stronger signal than Northern Europe.

-Line 189: The ERF of GHG2050 and AerGHG2050 seem quite different to me, AerGHG2050 seems much more like ALL2050.

-Line 268-270: But this is indeed what the scenario SSP1-1.9 assumes, negative emissions of carbon dioxide after 2050 (IPCC, 2021 SPM).

References:

IPCC, 2021: Summary for Policymakers. In: Climate Change 2021: The Physical Science Basis. Contribution of Working Group I to the Sixth Assessment Report of the Intergovernmental Panel on Climate Change [Masson-Delmotte, V., P. Zhai, A. Pirani, S.L. Connors, C. Péan, S. Berger, N. Caud, Y. Chen, L. Goldfarb, M.I. Gomis, M. Huang, K. Leitzell, E. Lonnoy, J.B.R. Matthews, T.K. Maycock, T. Waterfield, O. Yelekçi, R. Yu, and B. Zhou (eds.)]. In Press.

Reviewer #1:

The authors present the impacts of greenhouse gases, aerosols, and tropospheric O₃ on climate and extreme weather events under a carbon neutrality pathway. With the Community Earth System Model version 1 (CESM1), the authors find that the climate effects of aerosol reductions far outweigh those of GHGs and tropospheric O₃ under carbon neutrality, which reverses the knowledge that the changing GHGs dominate the future climate changes as predicted in the medium forcing pathway. This study addresses very important questions that how the climate and extreme weather events evolve under a carbon neutrality scenario and which external forcing contributes the most to the climate change. The paper presents novel ideas and analysis and the scientific methods and assumptions are valid and clearly outlined. Before this paper can be considered for publication, I have a few minor comments need to be addressed.

Reply: We thank the reviewer for the constructive comments and suggestions, which are very helpful for improving the clarity and reliability of the manuscript. Please see our point-by-point responses to your comments below.

1. It is interesting that the carbon neutrality scenario (SSP1-1.9) presents different contributions of aerosols on climate as compared to SSP2-4.5. The author can compare aerosol reductions under SSP1-1.9 with those under SSP2-4.5.

Reply: Thanks for your constructive and helpful suggestions. As we can see, the emission reductions of aerosols and their precursors are relatively weaker under SSP2-4.5 (Fig. R1, see below) compared to those under SSP1-1.9 (Fig. S6). For example, in 2100, SO₂ emissions will be reduced by less than 40% under SSP2-4.5 (Fig. R1a), but by more than 60% under SSP1-1.9 (Fig. S6b) in most subregions compared to 2020 levels. For BC and OC emissions, substantial differences in aerosol emissions between SSP2-4.5 (Fig. R1b&c) and SSP1-1.9 (Fig. S6c&d) are also seen.

Figure R1 Regional mean reductions in emissions (%) of SO₂ (a), BC (b) and OC (c) in 2050 and 2100, relative 2020, under SSP2-4.5 scenario.

2. According to the model simulations, GHGs have very slight impacts on temperatures and precipitations compared to aerosols toward carbon neutrality. It is the most significant

finding and reverses the current climate projection in other scenarios. Does each of the 3 model ensemble members exhibit the same result?

Reply: Yes, the three ensemble members show almost the same results. Figures R2-R7 demonstrate the surface air temperature and annual mean precipitation changes relative to the 2020 levels in each of the three ensembles of GHG2050, AerGHG2050, ALL2050 and ALL2100 simulations. The spatial patterns of air temperature and precipitations are similar among three ensemble members for each experiment, suggesting that the changes in surface air temperature and precipitations driven by anthropogenic emissions are robust among the three ensemble members.

Figure R2 Changes in annual mean surface temperature (°C) in first ensemble member of GHG2050 (a), AerGHG2050 (b), ALL2050 (c), and ALL2100 (d), relative to Baseline (2020) simulation.

Figure R3 Same as in Figure R2, but for results in the second ensemble member of GHG2050 (a), AerGHG2050 (b), ALL2050 (c), and ALL2100 (d) simulations.

Figure R4 Same as in Figure R2, but for results in the third ensemble member of GHG2050 (a), AerGHG2050 (b), ALL2050 (c), and ALL2100 (d) simulations.

Figure R5 Changes in annual mean precipitation (mm/day) in the first ensemble member of GHG2050 (a), AerGHG2050 (b), ALL2050 (c), and ALL2100 (d), relative to Baseline (2020) simulation.

Figure R6 Same as in Figure R5, but for results in the second ensemble member of GHG2050 (a), AerGHG2050 (b), ALL2050 (c), and ALL2100 (d) simulations.

Figure R7 Same as in Figure R5, but for results in the second ensemble member of GHG2050 (a), AerGHG2050 (b), ALL2050 (c), and ALL2100 (d) simulations.

3. I wonder why you selected to run the simulations using fixed forcing for 2020, 2050, and 2100 to get a quasi-equilibrium response. Some studies presented climate responses using the transient simulations. It is not necessary that the author should perform new simulation, but perhaps you should discuss the possible difference between transient and equilibrium responses.

Reply: Thanks for pointing this out. As already stressed in the main context: “Although for a long time, many numerical studies investigate the climate under the 1.5°C/2°C global warming as a transient phase, the 1.5 °C/2 °C global warming limits are set by the Paris agreement with quasi-stabilized scenarios⁴⁸. Recent studies have pointed out the substantial differences in climate response between transient and stabilized climates that the local climate will eventually warm much more under transient global warming than equilibrium global warming at the same temperature threshold, causing differences in

evapotranspiration, precipitation, and surface temperature^{49, 72}. Our study aims to understand the future climate effects of different anthropogenic forcings under quasi-equilibrium conditions for the development of climate change adaptation policies.” We have added more discussions on the potential differences between transient and quasi-equilibrium simulations in the revised manuscript: “The transient and equilibrium responses of both surface air temperature and precipitation to CO₂ increases are broadly similar in spatial patterns over most of the globe among different forcing scenarios in CESM1 simulations⁴⁷. If the transient simulations are performed, the patterns of temperature/precipitation changes in response to GHGs changes would be similar to those obtained through the current equilibrium simulations following SSP1-1.9, but the responses to GHGs changes would be even weaker in transient simulations as compared to equilibrium simulations⁴⁸, which would not alter the finding in this study that the climate effects of aerosol reductions far outweigh those of GHGs under carbon neutrality.”

4. Recently, Wang et al. (2023, IJOC) have analyzed influences of anthropogenic and natural forcings on future changes in precipitation projected by the CMIP6-DAMIP model. It would be better to cite this paper to enhance the result obtained from this manuscript (Wang, Y., T. Zhao*, L. Hua, X. Guan, C. Xu, and X. Chen, 2023: *Int. J. Climatol.*, doi: 10.1002/ioc.8064)

Reply: Thanks for your suggestion. We have cited this study in the revised manuscript as: “As part of the Coupled Model Intercomparison Project Phase 6 (CMIP6), the Detection and Attribution Model Intercomparison Project (DAMIP) has been conducted to estimate the individual contributions of various external forcings to the observed global and regional climate changes^{26, 27}”.

5. About the climate Extreme Indices, the humid heat waves are defined based on daily maximum wet-bulb temperature (TW). The question is how TW is calculated? Is it an output of the model?

Reply: Daily maximum wet-bulb temperature is obtained from surface pressure, air temperature and specific humidity referring to the method. We have made it clear in the revised manuscript: “Using the algorithm described in Davies-Jones⁷⁷, the daily maximum TW is calculated from the daily maximum air temperature, daily mean relative humidity, and daily mean surface pressure.”

6. Orders of figures needed to be adjusted. For example, the supporting figure starts from Figure S3.

Reply: Thanks for pointing this out. The orders of all figures have been updated.

7. L115: 154: Table S3 to Table S2

Reply: The orders of all tables are updated.

8. L170-178: Are there any published works to support the finding here?

Reply: Thanks for pointing this out. Xu et al. (2022) have analyzed influences of anthropogenic and natural forcings on historical and future changes in global land-surface air temperature using model simulations from the CMIP6-DAMIP model. And their results suggest that greenhouse gases contribute the most to projected changes in land-surface air temperature, which support our finding here. Thus, we have we have cited this work in the revised manuscript: “The results are consistent with a recent study that GHGs contribute the most to projected changes in land surface air temperature under SSP2-4.5 emission pathway³¹.”

9. L318: SSP119 to SSP1-1.9

Reply: Revised.

10. Missing y-axis labels in Fig. S6d.

Reply: Added.

11. Inconsistent fonts, e.g., Fig. S3.

Reply: Revised.

References:

26. Gillett, N.P. et al. The detection and attribution model intercomparison project (DAMIP v1. 0) contribution to CMIP6. *Geosci. Model Develop.* 9(10), 3685-3697 (2016).
27. Wang, Y., Zhao, T., Hua, L., Guan, X., Xu, C. & Chen, Q. Influence of anthropogenic and natural forcings on future changes in precipitation projected by the CMIP6–DAMIP models. *Inter. J. Clim.* (2023)
31. Xu, C., Zhao, T., Zhang, J., et al. Impacts of natural and anthropogenic forcings on historical and future changes in the global land-surface air temperature simulated by CMIP6–DAMIP[J] (2022).
47. Huang, D., Dai, A., & Zhu, J. (2020). Are the transient and equilibrium climate change patterns similar in response to increased CO₂? *J. Clim.* 33(18), 8003-8023.
48. You, Q., Jiang, Z., Yue, X., Guo, W., Liu, Y., Cao, J., ... & Zhai, P. Recent frontiers of climate changes in East Asia at global warming of 1.5° C and 2° C. *Npj Clim. Atmospheric Sci.*, 5(1), 80. (2022).
49. King, A. D., Lane, T. P, Henley, B. J. & Brown, J. R. Global and regional impacts differ between transient and equilibrium warmer worlds. *Nat. Clim. Change* 10(1), 42-47 (2020).
72. Julien, B., Naota, H., Jacob, S. & Hideo, S. Magnitude and robustness associated with the climate change impacts on global hydrological variables for transient and stabilized climate states. *Environ. Res. Lett.* 13(6), 064017 (2018).

77. Davies-Jones, R. An efficient and accurate method for computing the wet-bulb temperature along pseudoadiabats. *Mon. Weather Rev.* 136(7), 2764-2785 (2008).

Reviewer #2:

This study assesses individually the global impacts of changing GHGs, aerosols, and tropospheric O₃ levels following a carbon neutrality pathway on climate and extreme weather events by using the Community Earth System Model version 1 (CESM1). The author found that the global changes in climate and extremes by 2050 and 2100 will be considerably stronger due to the emission reductions in aerosols and precursors than those due to GHGs and tropospheric O₃ changes. The study suggests that substantial reductions in GHGs and tropospheric O₃ are necessary to reach the 1.5 °C warming target.

Reply: We thank the reviewer for the constructive comments and suggestions, which are very helpful for improving the clarity and reliability of the manuscript. Please see our point-by-point responses to your comments below.

1. The finding is interesting although expected as the CESM1 model has a strong (negative) aerosol radiative forcing at present-day. A reduction in aerosol emissions in the future (e.g., 2050) implies a strong warming. As only one model (which may have too strong aerosol radiative forcing) is used in this study, the authors may consider the inclusion of results from the other models. The uncertainty of aerosol radiative forcing in the CESM1 model and implications to the results of this study should be discussed.

Reply: Thank you for the suggestions. We agree with the reviewer that CESM1 model estimates a relatively strong negative aerosol radiative forcing, due to the large negative cloud scattering components. Zelinka et al. (2014) reported that effective radiative forcing (ERF) at top of atmosphere (TOA) due to aerosol changes at present-day (2000) relative to preindustrial (1860) in CESM1 surpass most of the other global climate models, following CSIRO-Mk3-6-0 and GFDL-CM3 models. To minimize the uncertainty due to the model dependency, we have now added the comparison of the changes in ERF due to aerosol changes in several global climate models by applying scaling factors characterizing the

model differences in estimating ERF and discussed the potential impacts of model uncertainty in estimating aerosol ERF in CESM1.

The global mean changes in ERFs driven by aerosol reductions in 2050 relative to 2020 in nine global climate models are given in Fig. S14. Though the multi-model average of aerosol-driven changes in ERF (1.93 W/m^2) is lower than that in CESM1 (2.26 W/m^2), it is still far beyond the global mean changes in ERF due to GHGs changes (0.01 W/m^2). Thus, the dominant role of aerosol reduction in affecting climate change following the carbon neutrality pathway as stressed in this work would not be changed with multi-model simulations.

We have accordingly added discussions on the model uncertainty of aerosol ERF in CESM1 in the revised manuscript (Line 345-355): “The interactions of anthropogenic aerosols with radiation and clouds are the largest source of uncertainty in the estimation of effective radiative forcing⁵². According to Zelinka et al.⁵¹, among nine global climate models, ERF at TOA due to aerosol changes at present-day (2000) relative to preindustrial (1860) levels in CESM1 surpasses many other climate models and the aerosol ERF in CESM1 is 1.17 times as much as the multi-model average. Here, the global mean changes in ERFs driven by aerosol reductions in 2050 relative to 2020 in the nine global climate models used in Zelinka et al.⁵¹ are estimated and given in Fig. S14. Though the multi-model average of aerosol-driven changes in ERF (1.93 W/m^2) is lower than that in CESM1 (2.26 W/m^2), it is still far beyond the global mean changes in ERF due to GHGs changes (0.01 W/m^2). Therefore, the main finding that aerosol reductions dominate climate changes toward carbon neutrality target would not be changed with multi-model simulations.”

Figure S14 Estimated changes in global average effective radiative forcing (ERF, W/m²) at the top of the atmosphere due to aerosol changes in 2050 relative to 2020 levels. ERF changes in other global climate models other than CESM1 are calculated by multiplying the changes in ERF in CESM1 by the ratios of historical ERF (2000 Vs. 1860) between CESM1 and the other GCMs derived from ref. 51.

2. The authors mentioned 12 global climate models in CMIP6 under SSP1-1.9 scenario and used the multi-model ensemble mean (Figure S11 and Table S5). It would be interesting to include these model results in the analyses.

Reply: Thanks for the suggestion. The multi-model results are used to supported the projected climate change in this study, but they lack the simulations for the contributions from individual forcings. We have now added more discussions on the model uncertainty of using CESM1 in the revised manuscript by referring to previous work (**Line 337-344**): “Moreover, large model uncertainties exist in projections of climate response to anthropogenic forcing, and CESM1 is relatively more sensitive to anthropogenic forcings. The equilibrium climate sensitivity in CESM1-CAM5 is 4.1°C⁵⁰, exceeding the upper tail of the very likely range of 2.5 °C to 4 °C obtained by multi-model simulations in CMIP6³⁹. CESM1 generally has a higher ERF due to aerosol-cloud interactions than other climate

models related to its aerosol-ice cloud interaction⁵¹, partly explaining the relatively stronger responses in temperature and precipitation to the anthropogenic forcings in ALL2050 than the ensemble means of multi-models in CMIP6 by 2050 (Fig. S11).”**And (Line 345-353):** “The interactions of anthropogenic aerosols with radiation and clouds are the largest source of uncertainty in the estimation of ERF⁵². According to Zelinka et al. ⁵¹, among nine global climate models, ERF at TOA due to aerosol changes at present-day (2000) relative to preindustrial (1860) levels in CESM1 surpasses many other climate models and the aerosol ERF in CESM1 is 1.17 times as much as the multi-model average. Here, the global mean changes in ERFs driven by aerosol reductions in 2050 relative to 2020 in the nine global climate models used in ref. 51 are estimated and given in Fig. S14. Though the multi-model average of aerosol-driven changes in ERF (1.93 W/m²) is lower than that in CESM1 (2.26 W/m²), it is still far beyond the global mean changes in ERF due to GHGs changes (0.01 W/m²).” And we have stressed in the revised manuscript that it would be interesting to involve more models **(Line 355-357):** “Still, it is of great significance for the community to conduct multi-model intercomparison of future climate change in response to individual anthropogenic forcings under carbon neutrality scenario.”

3. This study used the approach for the equilibrium climate state (time slides) rather than the transient climate change approach commonly used in the IPCC AR6. This may make the comparison with the DAMIP multi-model simulations (SSP245) ambiguously.

Reply: Thank you for pointing this out. Recent studies have pointed out the substantial differences in climate response between transient and stabilized climates that the local climate will eventually warm much more under transient global warming than equilibrium global warming at the same temperature threshold ^{49, 72}. The transient and equilibrium responses of both surface air temperature and precipitation to CO₂ increases are broadly similar in spatial patterns over most of the globe among different forcing scenarios in

CESM1 simulations⁴⁷. If the quasi-equilibrium simulations are performed, the patterns of temperature/precipitation changes in response to GHGs changes would be similar to those obtained the DAMIP multi-model simulations (transient simulations), but the responses would be even stronger in quasi-equilibrium simulations as compared to transient simulations⁴⁸, which would not alter the finding in this study that the climate effects of aerosol reductions far outweigh those of GHGs under carbon neutrality. And it reverses the knowledge that the changing GHGs dominate the future climate changes as predicted in the middle of the road pathway. We have now added these discussions in the revised manuscript.

4. Although the impacts of aerosol on temperature and precipitation are well studied, a discussion on the mechanisms of aerosol impacts on extreme weather events may be required to be included as this is the most interesting part of this study.

Reply: As suggested, we have now accordingly added discussions on the mechanisms of aerosol impacts on extreme weather events in the revised manuscript (**Line 310-322**): “The anthropogenic forcings including GHGs and aerosols are the main drivers of the past and future changes in the climate as well as the extremes, as extremes are parts of the climate system. In particular, in the future of carbon neutrality, reduced aerosols enhance the warming effect caused by GHGs. Changes in extremes at the global and regional scales are a direct consequence of the enhanced radiative forcing, and the associated global warming and/or its resultant increase in the water-holding capacity of the atmosphere based on the Clausius-Clapeyron relationship through thermodynamical processes³⁰. At global scale, the number of heatwave days and the length of heatwave seasons in various regions scale well with global mean temperatures^{39,40}, and changes in annual maximum one-day precipitation are proportional to global mean temperature changes^{41,42}. At the local and regional scales, aerosol concentrations changes are also a key factor in controlling and modulating

temperature and extremes⁴³, through the combined effects of the atmospheric energy balance, dynamical adjustment, and vertical structure of forcing^{44, 45}.”

The writing of this paper needs to be improved:

1. The abstract is wordy and should be much condensed.

Reply: Abstract has been rewritten to be concise.

2. The appearance of the first figure is Fig. S3 (line 123). This seems not in the right order of figures, e.g., the first figure appearing in the text should be Figure S1.

Reply: Corrected.

3. When reading the results sections, it is unclear that these results were obtained using the CESM1 model. CESM1 is mentioned in the abstract and then in the Method section. E.g., for the results in Lines 121-125, how and which model is used to derive them?

Reply: Thanks for pointing this out. This structure of manuscript has been updated for a clearer description.

4. Line 257, what are “HWF, HWD, and HWA”. They need to be defined.

Reply: Thanks for pointing this out. All the variables have been defined.

5. Line 281, it uses “Pretotal” while “Pretol” is used in Figure 4 caption.

Reply: Corrected.

References:

30. IPCC, 2021: Climate Change 2021: The Physical Science Basis. Contribution of Working Group I to the Sixth Assessment Report of the Intergovernmental Panel on Climate Change [Masson-Delmotte, V., P. Zhai, A. Pirani, S.L. Connors, C. Péan, S. Berger, N. Caud, Y. Chen, L. Goldfarb, M.I. Gomis, M. Huang, K. Leitzell, E. Lonnoy, J.B.R. Matthews, T.K. Maycock, T. Waterfield, O. Yelekçi, R. Yu, and B. Zhou (eds.)]. Cambridge University Press, Cambridge, United Kingdom and New York, NY, USA, 2391 pp. doi:10.1017/9781009157896.
49. Wartenburger, R., Hirschi, M., Donat, M. G., Greve, P., Pitman, A. J., and Seneviratne, S. I.. Changes in regional climate extremes as a function of global mean temperature: an interactive plotting framework. *Geosci. Model Dev. Discuss.*, 1–30. doi:10.5194/gmd-2017-33 (2017).
40. Sun, Y., Hu, T., & Zhang, X. Substantial increase in heat wave risks in China in a future warmer world. *Earth's Future*, 6(11), 1528-1538 (2018).
41. Kharin, V. V., Zwiers, F. W., Zhang, X., and Wehner, M.. Changes in temperature and precipitation extremes in the CMIP5 ensemble. *Clim. Change* 119, 345–357. doi:10.1007/s10584-013-0705-8 (2013).
42. Westra, S., Alexander, L. V., and Zwiers, F. W. Global increasing trends in annual maximum daily precipitation. *J. Clim.* 26, 3904–3918. doi:10.1175/JCLI-D-12-00502.1 (2013).
43. Dong, B., Sutton, R. T., and Shaffrey, L. Understanding the rapid summer warming and changes in temperature extremes since the mid-1990s over Western Europe. *Clim. Dyn.* 48, 1537–1554. doi:10.1007/s00382-016-3158-8 (2017).
44. Lin, L., Wang, Z., Xu, Y., and Fu, Q. Sensitivity of precipitation extremes to radiative forcing of greenhouse gases and aerosols. *Geophys. Res. Lett.* 43, 9860–9868. doi:10.1002/2016GL070869 (2016).

45. Lin, L., Wang, Z., Xu, Y., Fu, Q., and Dong, W.. Larger Sensitivity of Precipitation Extremes to Aerosol Than Greenhouse Gas Forcing in CMIP5 Models. *J. Geophys. Res. Atmos.* doi:10.1029/2018JD028821 (2018).
 47. Huang, D., Dai, A., & Zhu, J. Are the transient and equilibrium climate change patterns similar in response to increased CO₂? *J. Clim.* 33(18), 8003-8023 (2020).
 48. You, Q., Jiang, Z., Yue, X., Guo, W., Liu, Y., Cao, J., ... & Zhai, P. Recent frontiers of climate changes in East Asia at global warming of 1.5° C and 2° C. *Npj Clim. Atmospheric Sci.*, 5(1), 80. (2022)
 49. King, A. D., Lane, T. P., Henley, B. J. & Brown, J. R. Global and regional impacts differ between transient and equilibrium warmer worlds. *Nat. Clim. Change* 10(1), 42-47 (2020).
 50. Meehl, G. A., Washington, W. M., Arblaster, J. M., Hu, A., Teng, H., Kay, J. E., ... & Strand, W. G. Climate change projections in CESM1 (CAM5) compared to CCSM4. *Journal of Climate*, 26(17), 6287-6308 (2013).
 51. Zelinka, M. D., Andrews, T., Forster, P. M., & Taylor, K. E. Quantifying components of aerosol-cloud-radiation interactions in climate models. *Journal of Geophysical Research: Atmospheres*, 119(12), 7599-7615 (2014).
 52. Boucher, O. et al. Clouds and Aerosols. In: *Climate Change: The Physical Science Basis. Contribution of Working Group I to the Fifth Assessment Report of the Intergovernmental Panel on Climate Change* [Stocker, T.F., D. Qin, G.-K. Plattner, M. Tignor, S.K. Allen, J. Boschung, A. Nauels, Y. Xia, V. Bex, and P.M. Midgley (eds.)]. Cambridge University Press, Cambridge, United Kingdom and New York, NY, USA, pp. 571–657, doi:10.1017/cbo9781107415324.016 (2013).
 72. Julien, B., Naota, H., Jacob, S. & Hideo, S. Magnitude and robustness associated with the climate change impacts on global hydrological variables for transient and stabilized climate states. *Environ. Res. Lett.* 13(6), 064017 (2018).
-

Reviewer #3:

The study investigates the relative contribution on future climate from the different climate forcings: well-mixed GHGs, aerosols and precursors, and ozone following a carbon-neutrality scenario using the model CESM. The main results are not ground-breaking or surprising, given the wealth of research showing the future changes in climate due to strong aerosol reductions (many cited in the manuscript). In my opinion the main value of the study is that it demonstrates that under a carbon-neutrality scenario, aerosols largely dominate the future climate response, something that is still perhaps overlooked by the community and carries an important political message. The study has a well-defined scientific question, the methodology followed is clearly described and the manuscript is well written.

Reply: We thank the reviewer for the constructive comments and suggestions, which are very helpful for improving the clarity and reliability of the manuscript. Please see our point-by-point responses to your comments below.

General comments:

1. It seems like the CESM simulations show quite stronger responses in temperature and precipitation than the mean of 12 CMIP6 models under scenario SSP1-1.9 by 2050 (Figs 1 and 2 vs. Fig S11). If scenario SSP1-1.9 by definition limits warming to around 1.5 degrees based on a multi-model ensemble (See table SPM1 in IPCC, 2021: SPM), where does CESM stand in this ensemble? Is CESM more sensitive to aerosol reductions than the multi-model mean? I think there should be enough literature comparing CESM with other models to try and better understand where the model stands in terms of aerosol responses. Would the main conclusions be the same if the model was very insensitive to aerosol forcing? Or are these differences related to the semi-equilibrium response as opposed to the transient one?

Reply: Thank you for the very useful comments and suggestions. Yes, the CESM1 results should stand around the upper tail of the multi-model ensemble related to its strong aerosol ERF, despite the CMIP6 does not include CESM1 simulation. However, the strong sensitivity and equilibrium response should not influence the main results in this study. Please see the responses below.

1) For the strong sensitivity of CESM1 to anthropogenic aerosols, we have added the following clarifications in the updated manuscript (Line 337-344): “Moreover, large model uncertainties exist in projections of climate response to anthropogenic forcing, and CESM1 is relatively more sensitive to anthropogenic forcings. The equilibrium climate sensitivity in CESM1-CAM5 is 4.1°C^{50} , exceeding the upper tail of the very likely range of 2.5°C to 4°C obtained by multi-model simulations in CMIP6³⁰. CESM1 generally has a higher ERF due to aerosol-cloud interactions than other climate models related to its aerosol-ice cloud interaction⁵¹, partly explaining the relatively stronger responses in temperature and precipitation to the anthropogenic forcings in ALL2050 than the ensemble means of multi-models in CMIP6 by 2050 (Fig. S11).”and **Line (345-355):** “The interactions of anthropogenic aerosols with radiation and clouds are the largest source of uncertainty in the estimation of effective radiative forcing⁵². According to Zelinka et al. ⁵¹, among nine global climate models, ERF at TOA due to aerosol changes at present-day (2000) relative to preindustrial (1860) levels in CESM1 surpasses many other climate models and the aerosol ERF in CESM1 is 1.17 times as much as the multi-model average. Here, the global mean changes in ERFs driven by aerosol reductions in 2050 relative to 2020 in the nine global climate models used in Zelinka et al. ⁵¹ are estimated and given in Fig. S14. Though the multi-model average of aerosol-driven changes in ERF (1.93 W/m^2) is lower than that in CESM1 (2.26 W/m^2), it is still far beyond the global mean changes in ERF due to GHGs changes (0.01 W/m^2). Therefore, the main finding that aerosol reductions dominate climate changes toward carbon neutrality target would not be changed with multi-model simulations.”

Figure S14 Estimated changes in global average effective radiative forcing (ERF, W/m²) at the top of the atmosphere due to aerosol changes in 2050 relative to 2020 levels. ERF changes in other global climate models other than CESM1 are calculated by multiplying the changes in ERF in CESM1 by the ratios of historical ERF (2000 Vs. 1860) between CESM1 and the other GCMs derived from ref. 51.

2) **For the potential differences between equilibrium climate simulation and transient climate simulations, we have now added the following discussions in the revised manuscript:** “Recent studies have pointed out the substantial differences in climate response between transient and stabilized climates that the local climate will eventually warm much more under transient global warming than equilibrium global warming at the same temperature threshold” (Line 411-414), and “The transient and equilibrium responses of both surface air temperature and precipitation to CO₂ increases are broadly similar in spatial patterns over most of the globe among different forcing scenarios in CESM1 simulations⁴⁷. If the transient simulations are performed, the patterns of temperature/precipitation changes in response to GHGs changes would be similar to those obtained through the current equilibrium simulations following SSP1-1.9, but the

responses to GHGs changes would be even weaker in transient simulations as compared to equilibrium simulations⁴⁸, which would not alter the finding in this study that the climate effects of aerosol reductions far outweigh those of GHGs under carbon neutrality. (Line 326-333)”

2. The authors mention many and very valid limitations and uncertainties, but miss the most important one, which is model uncertainty. The qualitative comparison between Figs. 1-2 and Fig S11 seems insufficient to claim that CESM represents the real climate response to the studied forcings. This is not a particular argument against the model, the same could be said about any model. I do think that the authors need to clearly state this in the discussion and wherever relevant in the manuscript. Furthermore, this relative weakness could be turned into a powerful message: encourage the community to pursue in future multi-model detection and attribution assessments this type of best case, low carbon scenarios which may still carry important negative consequences.

Reply: Thank you for pointing this out. We agree with the reviewer that model uncertainty is important and we have now made it clearer in the manuscript and added discussions in the revised manuscript associated the model uncertainty:

1) **We have added discussions on the significance of model uncertainty (Line 337-344):** “Moreover, large model uncertainties exist in projections of climate response to anthropogenic forcing, and CESM1 is relatively more sensitive to anthropogenic forcings. The equilibrium climate sensitivity in CESM1-CAM5 is 4.1 °C⁵⁰, exceeding the upper tail of the very likely range of 2.5 °C to 4 °C obtained by multi-model simulations in CMIP6³⁰. CESM1 generally has a higher ERF due to aerosol-cloud interactions than other climate models related to its aerosol-ice cloud interaction⁵¹, partly explaining the relatively stronger responses in temperature and precipitation to

the anthropogenic forcings in ALL2050 than the ensemble means of multi-models in CMIP6 by 2050 (Fig. S11).”

- 2) **We have compared the effective radiative forcing (ERF) due to aerosol changes in CESM1 with several other global climate models and discussed the impacts of model uncertainty on the current results (Line 345-355):** “The interactions of anthropogenic aerosols with radiation and clouds are the largest source of uncertainty in the estimation of effective radiative forcing⁵². According to ref. 51, among nine global climate models, ERF at TOA due to aerosol changes at present-day (2000) relative to preindustrial (1860) levels in CESM1 surpasses many other climate models and the aerosol ERF in CESM1 is 1.17 times as much as the multi-model average. Here, the global mean changes in ERFs driven by aerosol reductions in 2050 relative to 2020 in the nine global climate models used in ref. 51 are estimated and given in Fig. S14. Though the multi-model average of aerosol-driven changes in ERF (1.93 W/m^2) is lower than that in CESM1 (2.26 W/m^2), it is still far beyond the global mean changes in ERF due to GHGs changes (0.01 W/m^2). Therefore, the main finding that aerosol reductions dominate climate changes toward carbon neutrality target would not be changed with multi-model simulations.”
- 3) **We have now emphasized the significance to conduct multi-model detection and attribution assessments toward carbon neutrality (Line 355-357):** “Still, it is of great significance for the community to conduct multi-model intercomparison of future climate change in response to individual anthropogenic forcings under carbon neutrality scenario.”

3. In addition, despite of apparent negative climate impacts of future aerosols emission reductions, major health benefits could be expected in regions with air pollution, something that in my opinion deserves to be highlighted in the manuscript. In fact, the warming due

to aerosol reductions is actually a committed warming due to rising long-lived GHGs, the aerosols are simply masking that warming. Acknowledging the benefits of anthropogenic aerosols reductions could help frame the message in a more balanced manner.

Reply: Thank you for this helpful and constructive suggestion. We have now added the benefits of anthropogenic aerosols reductions and stated the committed warming of aerosol reductions due to rising long-lived GHGs in the revised manuscript (**Line 373-377**): “It should be noted that the warming due to aerosol reductions toward carbon neutrality are actually a committed warming due to rising long-lived GHGs, and the aerosol reductions simply unmask that GHGs-induced warming. In addition, despite of negative climate impacts of future aerosols emission reductions, the benefits of carbon neutrality on air quality and global health burden are worthy of attention^{57, 58}”

4. The fact that the climate response is stronger by 2100 than by 2050 means that at least according to CESM, not even negative carbon dioxide emissions after 2050 following the SSP1-1.9 would be enough to compensate for the further reductions of aerosols after 2050? If so, this can be an important message to convey.

Reply: Thanks for pointing this out. We have added message in the revised manuscript (**Line 160-166**): “By further controlling anthropogenic emissions by the end of the century, the air temperature increases are further enhanced in the ALL2100 simulations especially in the mid-to-high latitudes, compared to those in ALL2050 (Fig. 1d), which is primarily contributed by the warming effects associated with more significant reductions in aerosols. It indicates that even negative CO₂ emissions after 2050 following the SSP1-1.9³⁰ would not be enough to compensate for the excessive warming caused by further reductions of aerosols after 2050.”

Specific comments:

-Lines 79-81: Is this sentence necessary? It seems a bit isolated from the context. Why Portugal?

Reply: The sentence has been removed.

-Line 146: Missing y-axis label in Fig S6d.

Reply: Added.

-Line 188: Seems like South America has a stronger signal than Northern Europe.

Reply: Thanks for pointing this out. This sentence has been corrected: “Compared to Baseline, the changes in ERF at TOA due to changes in GHG concentrations are weakly positive, with a maximum of 0.5-1.0 W/m² over North America, South America, and West Africa (Fig. S8a).”

-Line 189: The ERF of GHG2050 and AerGHG2050 seem quite different to me, AerGHG2050 seems much more like ALL2050.

Reply: This sentence has been corrected in the revised manuscript: “The TOA ERF changes between AerGHG2050 and Baseline are much larger, reaching 4.5 W/m² over East Asia and 2.5 W/m² over Northern Europe, North America, and East Africa (Fig. S8b).”

-Line 268-270: But this is indeed what the scenario SSP1-1.9 assumes, negative emissions of carbon dioxide after 2050 (IPCC, 2021 SPM).

Reply: Thanks for pointing this out. We have updated this statement and cited IPCC (2021): “Therefore, substantial reductions in O₃ precursors and GHGs emissions would have to be implemented to counteract the harmful climate consequences of future aerosol decline, which is also highlighted in SSP1-1.9 that CO₂ emissions would decline to net zero around 2050, followed by net negative CO₂ emissions³⁰”.

References:

30. IPCC, 2021: Climate Change 2021: The Physical Science Basis. Contribution of Working Group I to the Sixth Assessment Report of the Intergovernmental Panel on Climate Change [Masson-

Delmotte, V., P. Zhai, A. Pirani, S.L. Connors, C. Péan, S. Berger, N. Caud, Y. Chen, L. Goldfarb, M.I. Gomis, M. Huang, K. Leitzell, E. Lonnoy, J.B.R. Matthews, T.K. Maycock, T. Waterfield, O. Yelekçi, R. Yu, and B. Zhou (eds.)]. Cambridge University Press, Cambridge, United Kingdom and New York, NY, USA, 2391 pp. doi:10.1017/9781009157896.

47. Huang, D., Dai, A., & Zhu, J. Are the transient and equilibrium climate change patterns similar in response to increased CO₂? *J. Clim.* 33(18), 8003-8023 (2020).
48. You, Q., Jiang, Z., Yue, X., Guo, W., Liu, Y., Cao, J., ... & Zhai, P. Recent frontiers of climate changes in East Asia at global warming of 1.5° C and 2° C. *Npj Clim. Atmospheric Sci.*, 5(1), 80. (2022)
50. Meehl, G. A., Washington, W. M., Arblaster, J. M., Hu, A., Teng, H., Kay, J. E., ... & Strand, W. G. Climate change projections in CESM1 (CAM5) compared to CCSM4. *Journal of Climate*, 26(17), 6287-6308 (2013).
51. Zelinka, M. D., Andrews, T., Forster, P. M., & Taylor, K. E. Quantifying components of aerosol-cloud-radiation interactions in climate models. *Journal of Geophysical Research: Atmospheres*, 119(12), 7599-7615 (2014).
52. Boucher, O. et al. Clouds and Aerosols. In: *Climate Change: The Physical Science Basis. Contribution of Working Group I to the Fifth Assessment Report of the Intergovernmental Panel on Climate Change* [Stocker, T.F., D. Qin, G.-K. Plattner, M. Tignor, S.K. Allen, J. Boschung, A. Nauels, Y. Xia, V. Bex, and P.M. Midgley (eds.)]. Cambridge University Press, Cambridge, United Kingdom and New York, NY, USA, pp. 571–657, doi:10.1017/cbo9781107415324.016 (2013).
57. Li, H., Yang, Y., Wang, H., Wang, P., Yue, X., & Liao, H. Projected Aerosol Changes Driven by Emissions and Climate Change Using a Machine Learning Method. *Environmental Science & Technology*, 56(7), 3884-3893 (2022).
58. Yang, H., Huang, X., Westervelt, D. M., Horowitz, L., & Peng, W. Socio-demographic factors shaping the future global health burden from air pollution. *Nature Sustainability*, 6(1), 58-68. (2023).

REVIEWER COMMENTS

Reviewer #1 (Remarks to the Author):

As I am satisfied with the revised version of manuscript, I recommend for acceptance of this manuscript.

Reviewer #2 (Remarks to the Author):

The authors have addressed many of my comments. I appreciate their efforts. However, there are still a few comments not fully addressed or there are some place where are not very clearly elucidated, particularly regarding the aerosol ERF.

1. The authors used the nine global climate models in Zelinka et al., (Line 345-355) "...ERF at TOA due to aerosol changes at present-day (2000) relative to preindustrial (1860) levels in CESM1 surpasses many other climate models and the aerosol ERF in CESM1 is 1.17 times as much as the multi-model average".

How do you get this 1.17 number? Please use the absolute value of the difference.

Aerosol ERF from preindustrial to present-day in CESM1 is about -1.8 W m^{-2} . This is substantially higher than the other climate models assessed in IPCC AR5 and AR6. Is there a reason why the authors used the models in Zelinka et al.? Ideally the authors should refer to the models in IPCC ARs, not the models in Zelinka et al.

2. "Though the multimodel average of aerosol-driven changes in ERF (1.93 W/m^2) is lower than that in CESM1 (2.26 W/m^2), it is still far beyond the global mean changes in ERF due to GHGs changes (0.01 W/m^2)." This 0.01 W/m^2 of ERF due to GHGs changes in 2050 relative to 2020 is very small. Where do you derive this number?

3. The caption of Figure S14. Should it be "by multiplying the changes in ERF in CESM1 by the ratios of historical ERF (2000 Vs. 1860) between the other GCMs derived from ref. 51 and CESM1"?

4. The authors replied to my major comment #2, that the multi-model results lack the simulations for the contributions from individual forcings. Then can the authors compare the multi-model results with the ALL2050 and ALL2100 simulations from CESM1?

5. In line 337-344: "CESM1 generally has a higher ERF due to aerosol-cloud interactions than other climate models related to its aerosol-ice cloud interaction". I don't agree. The higher ERF due to aerosol-cloud interactions in CESM1 should be due to the aerosol-warm cloud interactions.

6. There is a confusion when comparing the transient and equilibrium responses: "that the local climate

will eventually warm much more under transient global warming than equilibrium global warming at the same temperature threshold 49, 72.” And “but the responses would be even stronger in quasi-equilibrium simulations as compared to transient simulations⁴⁸”. Please explain more clearly.

Reviewer #3 (Remarks to the Author):

The authors have addressed all my comments and changed the manuscript accordingly. I have no further comments.

Reviewer #2:

The authors have addressed many of my comments. I appreciate their efforts. However, there are still a few comments not fully addressed or there are some place where are not very clearly elucidated, particularly regarding the aerosol ERF.

Reply: We thank the reviewer for the constructive comments and suggestions, which are very helpful for improving the clarity and reliability of the manuscript. Please see our point-by-point responses to your comments below.

1. The authors used the nine global climate models in Zelinka et al., (Line 345-355) “...ERF at TOA due to aerosol changes at present-day (2000) relative to preindustrial (1860) levels in CESM1 surpasses many other climate models and the aerosol ERF in CESM1 is 1.17 times as much as the multi-model average”. How do you get this 1.17 number? Please use the absolute value of the difference.

Aerosol ERF from preindustrial to present-day in CESM1 is about -1.8 W m^{-2} . This is substantially higher than the other climate models assessed in IPCC AR5 and AR6. Is there a reason why the authors used the models in Zelinka et al.? Ideally the authors should refer to the models in IPCC ARs, not the models in Zelinka et al.

Reply: Thank you for pointing out the inappropriate descriptions. According to Zelinka et al (2014), aerosol ERF at TOA in present-day (2000) relative to the preindustrial (1860) level is -1.37 W/m^2 in CESM1 and -1.17 W/m^2 averaged over nine global climate models. The number 1.17 is the ratio between -1.37 W/m^2 and -1.17 W/m^2 . All the modeling results of Zelinka et al (2014) have been adopted in both IPCC AR5 and AR6 (Figure 8.23 in IPCC AR5 and Table 7.6 in IPCC AR6 as shown below).

In the revised manuscript, we have modified the descriptions as: “Aerosol ERF at TOA in present-day (2000) relative to the preindustrial (1860) level simulated by CESM1 is -1.37 W/m^2 ⁵¹ and -1.44 W/m^2 with an additional 5% applied to account for land-surface

cooling ⁵³, relative higher than the multi-model averages of $-1.23 \pm 0.48 \text{ W/m}^2$ in the Coupled Model Intercomparison Project Phase 5 (CMIP5) over 1860–2000 and $-1.11 \pm 0.38 \text{ W/m}^2$ in the CMIP6 over 1850–2014 ³⁰.”

Table 7.6 | Present-day effective radiative forcing (ERF) due to changes in aerosol–radiation interactions (ERFari) and changes in aerosol–cloud interactions (ERFaci), and total aerosol ERF (ERFari+aci) from GCM CMIP6 (2014 relative to 1850; Smith et al., 2020b and later model results) and CMIP5 (year 2000 relative to 1860; Zelinka et al., 2014). CMIP6 results are simulated as part of RFMIP (Pincus et al., 2016). An additional 5% is applied to the CMIP5 and CMIP6 model results to account for land-surface cooling (Figure 7.4; Smith et al., 2020a).

Models	ERFari (W m ⁻²)	ERFaci (W m ⁻²)	ERFari+aci (W m ⁻²)
ACCESS-CM2	-0.24	-0.93	-1.17
ACCESS-ESM1-5	-0.07	-1.19	-1.25
BCC-ESM1	-0.79	-0.69	-1.48
CanESM5	-0.02	-1.09	-1.11
CESM2	+0.15	-1.65	-1.50
CNRM-CM6-1	-0.28	-0.86	-1.14
CNRM-ESM2-1	-0.15	-0.64	-0.79
EC-Earth3	-0.39	-0.50	-0.89
GFDL-CM4	-0.12	-0.72	-0.84
GFDL-ESM4	-0.06	-0.84	-0.90
GISS-E2-1-G (physics_version=1)	-0.55	-0.81	-1.36
GISS-E2-1-G (physics_version=3)	-0.64	-0.39	-1.02
HadGEM3-GC31-LL	-0.29	-0.87	-1.17
IPSL-CM6A-LR	-0.39	-0.29	-0.68
IPSL-CM6A-LR-INCA	-0.45	-0.35	-0.80
MIROC6	-0.22	-0.77	-0.99
MPI-ESM-1-2-HAM	+0.10	-1.40	-1.31
MRI-ESM2-0	-0.48	-0.74	-1.22
NorESM2-LM	-0.15	-1.08	-1.23
NorESM2-MM	-0.03	-1.26	-1.29
UKESM1-0-LL	-0.20	-0.99	-1.19
CMIP6 average and 5–95% confidence range (2014 relative to 1850)	-0.25 ± 0.40	-0.86 ± 0.57	-1.11 ± 0.38
CMIP5 average and 5–95% confidence range (2000 relative to 1860)	-0.27 ± 0.35	-0.96 ± 0.55	-1.23 ± 0.48

2. “Though the multimodel average of aerosol-driven changes in ERF is lower than that in CESM1, it is still far beyond the global mean changes in ERF due to GHGs changes (0.01 W/m²).” This 0.01 W/m² of ERF due to GHGs changes in 2050 relative to 2020 is very small. Where do you derive this number?

Reply:

The 0.01 W/m² is obtained by averaging the global ERF due to GHGs changes in 2050 relative to 2020 (Figure S8a). As shown in Table S1, the global mean concentrations of CO₂ would increase from 414.04 ppm to 437.48 ppm during 2020 to 2050, but the CH₄ would decrease from 1884.13 ppb to 1429.12 ppb, partly compensating the CO₂ increases. Therefore, the ERF due to GHGs changes in 2050 relative to 2020 should be small. Accordingly, we have made it more clearly in the updated manuscript: “it is still far beyond the global averaged ERF due to GHGs changes (0.01 W/m²), as shown in Figure S8a”.

Figure S8 Changes in effective radiative forcing (ERF) at top of atmosphere (W/m^2) in GHG2050 (a), AerGHG2050(b), ALL2050 (c), and ALL2100 (d), relative to Baseline (2020). Regional mean changes in ERF at top of atmosphere in 2050 attributed to GHGs, aerosols and tropospheric O_3 changes over 21 subregions (e). The stippled areas indicate statistical significance with 95% confidence from a two-tailed Student's t-test.

3. The caption of Figure S14. Should it be “by multiplying the changes in ERF in CESM1 by the ratios of historical ERF (2000 Vs. 1860) between the other GCMs derived from ref. 51 and CESM1”?

Reply: Thanks. We have modified the caption of Figure S14 accordingly.

4. The authors replied to my major comment #2, that the multi-model results lack the simulations for the contributions from individual forcings. Then can the authors compare the multi-model results with the ALL2050 and ALL2100 simulations from CESM1?

Reply:

Thanks for this suggestion. The climate changes in 2050 relative to 2020 simulated in this study are compared with those from 13 global climate models in CMIP6 under SSP1-1.9 scenario, as the follows: “It is worth noting that the spatial pattern of changes in surface air temperature (Figure 1c) and precipitation (Figure 2c) in ALL2050 simulated by CESM1 are similar to those projected by the multi-model ensemble mean of 13 global climate models in CMIP6 under SSP1-1.9 scenario (Figure S11) with statistical significant spatial correlations above 0.8 for temperature change and 0.4 for precipitation change. Specifically, large increases in air temperature locate over mid-to-high latitudes of the Northern Hemisphere and increased rainfall anomalies mainly over the north of the equator, implying the crucial role of changing anthropogenic emissions in the future climate. It should be noted that the magnitudes of changes in air temperatures and precipitations in CESM1 are higher than those in multi-model results under SSP1-1.9. To be more specific,

the global mean changes in air temperatures and precipitation in 2050 relative to 2020 simulated by CESM1 are 0.85 °C and 0.07 mm/day, respectively, which are higher than the changes predicted by CMIP6 averages (0.60 ± 0.50 °C and 0.04 ± 0.24 mm/day) related to the larger ERFs due to aerosol reductions simulated in CESM1. Moreover, the global mean future changes in surface air temperatures and precipitations in 2100 relative to 2020 levels are 0.92 °C and 0.10 mm/day in CESM1, also exceeding the CMIP6 averages (0.40 ± 0.60 °C and 0.04 ± 0.24 mm/day).”

Figure S11 (a, b) Changes in annual mean surface air temperature (°C) and precipitation (mm/day), (c, d) Standard deviation of changes in surface air temperature and precipitation for the 13 CMIP6 models.

5. In line 337-344: “CESM1 generally has a higher ERF due to aerosol-cloud interactions than other climate models related to its aerosol-ice cloud interaction”. I don’t agree. The higher ERF due to aerosol-cloud interactions in CESM1 should be due to the aerosol-warm cloud interactions.

Reply: Thank you for the correction. We have deleted this statement and revised the description as: “CESM1 generally has a higher ERF due to aerosol-cloud interactions than other climate models ⁵¹, partly explaining the relatively stronger responses in temperature and precipitation to the anthropogenic forcings in ALL2050 than the ensemble means of multi-models in CMIP6 by 2050.”

6. There is a confusion when comparing the transient and equilibrium responses: “that the local climate will eventually warm much more under transient global warming than equilibrium global warming at the same temperature threshold ^{49, 72}.” And “but the responses would be even stronger in quasi-equilibrium simulations as compared to transient simulations ⁴⁸”. Please explain more clearly.

Reply: Thanks for pointing this out.

- ⑩ In the **Experimental design** section, we have revised the statement as “Recent studies have revealed substantial differences in climate responses between transient and stabilized climates and suggested that it would be valuable to design experiments in meeting the needs of decision-makers given that the Paris Agreement is implicitly targeting stabilized global warming levels^{49, 73}.”
- ⑩ For **the statement in the discussion section** “but the responses would be even stronger in quasi-equilibrium simulations as compared to transient simulations ⁴⁸” is indicated by previous works that in response to a given changes in CO₂ concentrations, such as a quadrupling of the preindustrial CO₂ (4×CO₂) or a doubling of the preindustrial CO₂ (2×CO₂), the global mean surface air temperature get warmer in the quasi-equilibrium experiment than the transient one (Huang et al., 2020; You et al., 2022).

References:

- Huang, D., Dai, A., & Zhu, J. Are the transient and equilibrium climate change patterns similar in response to increased CO₂? *J. Clim.* 33(18), 8003-8023 (2020).
- You, Q., Jiang, Z., Yue, X., Guo, W., Liu, Y., Cao, J., ... & Zhai, P. Recent frontiers of climate changes in East Asia at global warming of 1.5° C and 2° C. *Npj Clim. Atmospheric Sci.*, 5(1), 80. (2022)

REVIEWERS' COMMENTS

Reviewer #2 (Remarks to the Author):

The authors have sufficiently addressed my comments in the last round. I found one typo and one minor error in their revised manuscript.

1. in revised text: "relative higher than the multi-model averages of -1.23 ± 0.48 W/m² in the Coupled Model Intercomparison Project Phase 5 (CMIP5)". Here "relative" should be "relatively".

2. Figure S11 caption. There is an error in the order of (a)-(d). (a) and (b) are the mean and standard deviation of temperature. (c) and (d) are for precipitation.

Reviewer #2:

The authors have sufficiently addressed my comments in the last round. I found one typo and one minor error in their revised manuscript.

1. in revised text: "relative higher than the multi-model averages of -1.23 ± 0.48 W/m² in the Coupled Model Intercomparison Project Phase 5 (CMIP5)". Here "relative" should be "relatively".

Reply: Changed.

2. Figure S11 caption. There is an error in the order of (a)-(d).

(a) and (b) are the mean and standard deviation of temperature. (c) and (d) are for precipitation.

Reply: Changed.